# Towards a unified model of naive T cell dynamics across the lifespan

Sanket Rane[1,2†], Thea Hogan[3†], Edward Lee[4], Benedict Seddon[3]*, Andrew J Yates[1]*

[1]Department of Pathology and Cell Biology, Columbia University Irving Medical Center, New York, United States; [2]Irving Institute for Cancer Dynamics, Columbia University, New York, United States; [3]Institute of Immunity and Transplantation, Division of Infection and Immunity, UCL, Royal Free Hospital, London, United Kingdom; [4]Department of Laboratory Medicine, Yale University School of Medicine, New Haven, United States

**Abstract** Naive CD4 and CD8 T cells are cornerstones of adaptive immunity, but the dynamics of their establishment early in life and how their kinetics change as they mature following release from the thymus are poorly understood. Further, due to the diverse signals implicated in naive T cell survival, it has been a long-held and conceptually attractive view that they are sustained by active homeostatic control as thymic activity wanes. Here we use multiple modelling and experimental approaches to identify a unified model of naive CD4 and CD8 T cell population dynamics in mice, across their lifespan. We infer that both subsets divide rarely, and progressively increase their survival capacity with cell age. Strikingly, this simple model is able to describe naive CD4 T cell dynamics throughout life. In contrast, we find that newly generated naive CD8 T cells are lost more rapidly during the first 3–4 weeks of life, likely due to increased recruitment into memory. We find no evidence for elevated division rates in neonates, or for feedback regulation of naive T cell numbers at any age. We show how confronting mathematical models with diverse datasets can reveal a quantitative and remarkably simple picture of naive T cell dynamics in mice from birth into old age.

**\*For correspondence:**
benedict.seddon@ucl.ac.uk (BS); andrew.yates@columbia.edu (AJY)

†These authors contributed equally to this work

**Competing interest:** The authors declare that no competing interests exist.

## Editor's evaluation

This paper challenges the widely held view that the number of naive T cells in our body is regulated through homeostatic feedback mechanisms, meaning that cells divide more frequently – or live longer – when cell numbers are low. The arguments in favor of this homeostatic regulation rely on cross-sectional data, which fail to distinguish between the effects of host age, cell age, and cell numbers, each of which separately may influence the dynamics of cells. Using a set of mathematical models and experimental data sets, this paper manages to tear these factors apart and reports that in mice there is no feedback regulation of naive T cell numbers.

## Introduction

Lifelong and comprehensive adaptive immunity depends upon generating naive CD4 and CD8 T cell populations with diverse repertoires of T cell receptors (TCRs). These must be established rapidly from birth and then maintained throughout life. In mice, the number of circulating naive T cells grows from tens of thousands at birth to tens of millions in several weeks, peaking at around 2 months of age (*Scollay et al., 1980*; *den Braber et al., 2012*) and waning thereafter. Quantifying the relative contributions of thymic influx, and of loss and self-renewal across the lifespan, processes that either boost

or preserve diversity, will therefore help us understand at a mechanistic level how the TCR repertoire is generated and evolves as an individual ages.

The consensus view is that, in adult mice, naive T cells have a mean lifespan of several weeks but a mean interdivision time of several years (*den Braber et al., 2012*; *Hogan et al., 2015*). This difference in timescales leads to the conclusion that, in mice, most naive T cells never divide and that their numbers are sustained largely by thymic export, which in adult mice contributes 1–2% of the peripheral pool size per day (*Egerton et al., 1990*; *Graziano et al., 1998*; *Scollay et al., 1980*; *den Braber et al., 2012*; *Hogan et al., 2015*). However, the dynamics of the naive pool may be radically different early in life, and it is unclear whether the rules that govern naive T cell dynamics in adults are the same in neonates. Indeed, there is considerable evidence that this is not the case. First, studies suggest that neonatal mice are lymphopenic, a state which, when artificially induced, supports the rapid expansion of newly introduced T cells through a mechanism referred to as lymphopenia-induced proliferation (LIP) (*Rocha et al., 1989*; *Almeida et al., 2001*; *Yates et al., 2008*; *Houston et al., 2011*; *Hogan et al., 2013*), and T cells transferred to healthy neonatal mice undergo cell divisions not observed in adult recipients (*Min et al., 2003*; *Le Campion et al., 2002*). However, the early establishment of naive compartments is still heavily reliant upon thymic output, since depletion of thymocytes in 2-week-old mice drives a rapid and transient 50–70% reduction of peripheral CD4 and CD8 T cell numbers (*Dzierzak et al., 1993*).

Second, memory T cell compartments are rapidly established in neonatal mice, which derive from the activation of naive T cells. For instance, we have shown that the rate of generation of memory CD4 T cells is elevated early in life, at levels influenced by the antigenic content of the environment (*Hogan et al., 2019*). This result suggests that high *per capita* rates of activation upon first exposure to environmental stimuli may increase the apparent rate of loss of naive CD4 T cells in neonatal mice. One might expect a similar process to occur with naive CD8 T cells, with substantial numbers of so-called 'virtual' memory CD8 T cells generated from naive T cells in the periphery soon after birth (*Akue et al., 2012*, *Smith et al., 2018*). Together, these observations suggest that the average residence times of naive T cells differ in neonates and adults.

Third, the post-thymic age of cells in neonates is inevitably more restricted than in adults. Following the dynamic period of their establishment, there is evidence that naive T cells do not die or self-renew at constant rates but continue to respond or adapt to the host environment (*Houston et al., 2008*). Recent thymic emigrants (RTE) are functionally distinct from mature T cells (*Adkins, 1999*; *Wang et al., 2016*), may be lost at a higher rate than mature naive T cells under healthy conditions (*Berzins et al., 1998*; *Berzins et al., 1999*; *Houston et al., 2011*; *van Hoeven et al., 2017*), and respond differently to mature naive cells under lymphopenia (*Houston et al., 2011*). In the early weeks of life, all naive T cells are effectively RTE. Phenotypic markers of RTE are poorly defined, however, and so without a strict definition of 'recent' it is difficult to reach a consensus description of their kinetics. It may be more appropriate to view maturation as a continuum of states, and indeed the net loss rates (the balance of loss and self-renewal) of both naive CD4 (*Rane et al., 2018*) and CD8 (*Rane et al., 2018*; *Reynaldi et al., 2019*) T cells in mice appear to fall smoothly with a cell's post-thymic age, a process we have referred to as adaptation (*Rane et al., 2018*). Such behaviour will lead to increasing heterogeneity in the kinetics of naive T cells over time, as the population's age-distribution broadens, and may also contribute to skewing of the TCR repertoire, through a 'first-in, last-out' dynamic in which older naive T cells become progressively fitter than newer immigrants (*Hogan et al., 2015*). A conceptually similar model, in which naive T cells accrue fitness with their age through a sequence of stochastic mutation events, has been used to explain the reduced diversity of naive CD4 T cells in elderly humans (*Johnson et al., 2012*).

Taken together, these results indicate that cell numbers, host age and cell age may all influence naive T cell dynamics to varying degrees. When dealing with cross-sectional observations of cell populations, these effects may be difficult to distinguish. For example, the progressive decrease in the population-average loss rate of naive T cells observed in thymectomised mice (*den Braber et al., 2012*) may not derive from reduced competition, as was suggested, but may also be explained by adaptation or selective effects (*Rane et al., 2018*). It is also possible that elevated loss rates of naive T cells early in life may not be an effect of the neonatal environment *per se*, but just a consequence of the nascent naive T cell pool being comprised almost entirely of RTE with intrinsically shorter residence times than mature cells. These uncertainties invite the use

of mathematical models to distinguish different descriptions of naive T cell population dynamics from birth into old age.

Here, we combine model selection tools with data from multiple distinct experimental systems to investigate the rules governing naive T cell maintenance across the full lifespan of the mouse. We used an established bone marrow chimera system to specifically measure and model production, division and turnover of naive T cells in adult mice. We then used an out-of-sample prediction approach to test and refine these models in the settings of the establishment of the naive T cell compartments in neonates, and – using a unique Rag/Ki67 reporter mouse model – characterising the dynamics of RTE and mature naive T cells. We find that naive CD4 T cells appear to follow consistent rules of behaviour throughout the mouse lifespan, dividing very rarely and with a progressive increase in survival capacity with cell age, with no evidence for altered behaviour in neonates. Naive CD8 T cells behave similarly, but with an additional, increased rate of loss during the first few weeks of life that may reflect high levels of recruitment into early memory populations. These models are able to explain diverse observations and present a remarkably simple picture in which naive T cells appear to be passively maintained throughout life, with gradually extending lifespans that compensate in part from the decline in thymic output, and with no evidence for feedback regulation of cell numbers.

## Results
### Naive CD4 and CD8 T cells divide very rarely in adult mice and expected lifespans increase with cell age

Reports from our group and others (*Hogan et al., 2015*; *Rane et al., 2018*; *Reynaldi et al., 2019*; *Mold et al., 2019*) show that the dynamics of naive CD4 and CD8 T cells in adult mice and humans depend on cell age, defined to be time since they (or their ancestor, if they have divided) were released from the thymus. All these studies found that the net loss rate, which is the balance of their rate of loss through death or differentiation, and self-renewal through division, decreases gradually with cell age for both subsets. It is unknown whether these adaptations modulate the processes that regulate their survival, or their ability to self-renew.

To address this question, we used a well-established system that we have developed to quantify lymphocyte dynamics at steady state in healthy mice (*Hogan et al., 2017*), with the addition of detailed measurements of cell proliferation activity throughout. Briefly, hematopoietic stem cells (HSCs) in the bone marrow (BM) are partially and specifically depleted by optimised doses of the transplant conditioning drug busulfan, and reconstituted with T- and B-cell-depleted BM from congenic donor mice. Chimerism rapidly stabilises among progenitors in the bone marrow (*Verheijen et al., 2020*) and thymus (*Hogan et al., 2015*) and is maintained for the lifetime of the mouse (*Figure 1A*). The host's peripheral lymphocyte populations are unperturbed by treatment, and as donor T cells develop they progressively replace host T cells in the periphery through natural turnover. This system allows us to estimate the rates of influx into different lymphocyte populations and the net loss rates of cells within them; identify subpopulations with different rates of turnover; and infer whether and how these dynamics vary with host and/or cell age (*Hogan et al., 2015*; *Gossel et al., 2017*; *Verheijen et al., 2020*). Here, we generated a cohort of busulfan chimeric mice who underwent bone marrow transplant (BMT) between 7 and 25 weeks of age. At different times post-BMT, we enumerated host and donor-derived thymocyte subsets and peripheral naive T cells from spleen and lymph nodes (see *Figure 1—figure supplement 1* for the flow cytometric gating strategy). We began by normalising the chimerism (fraction donor) within naive CD4 and CD8 T cells to that of DP1 thymocytes to remove the effect of variation across mice in the stable level of bone-marrow chimerism. This normalised donor fraction ($f_d$) will approach 1 within a population if it turns over completely – that is, if its donor:host composition equilibrates to that of its precursor. Saturation at $f_d < 1$ implies incomplete replacement (*Figure 1A*), which can occur either through waning influx from the precursor population, or if older (host) cells persist longer than new (donor) cells, on average, implying cell-age effects on turnover or self-renewal. Previously, we observed incomplete replacement of both naive CD4 and CD8 T cells in adult busulfan chimeric mice (*Hogan et al., 2015*), and excluded the possibility that this shortfall derived from the natural involution of the thymus, leading us to infer that the net loss rates of both subsets increase with cell age (*Hogan et al., 2015*; *Rane et al., 2018*). For the present study, we also used concurrent measurements of Ki67, a

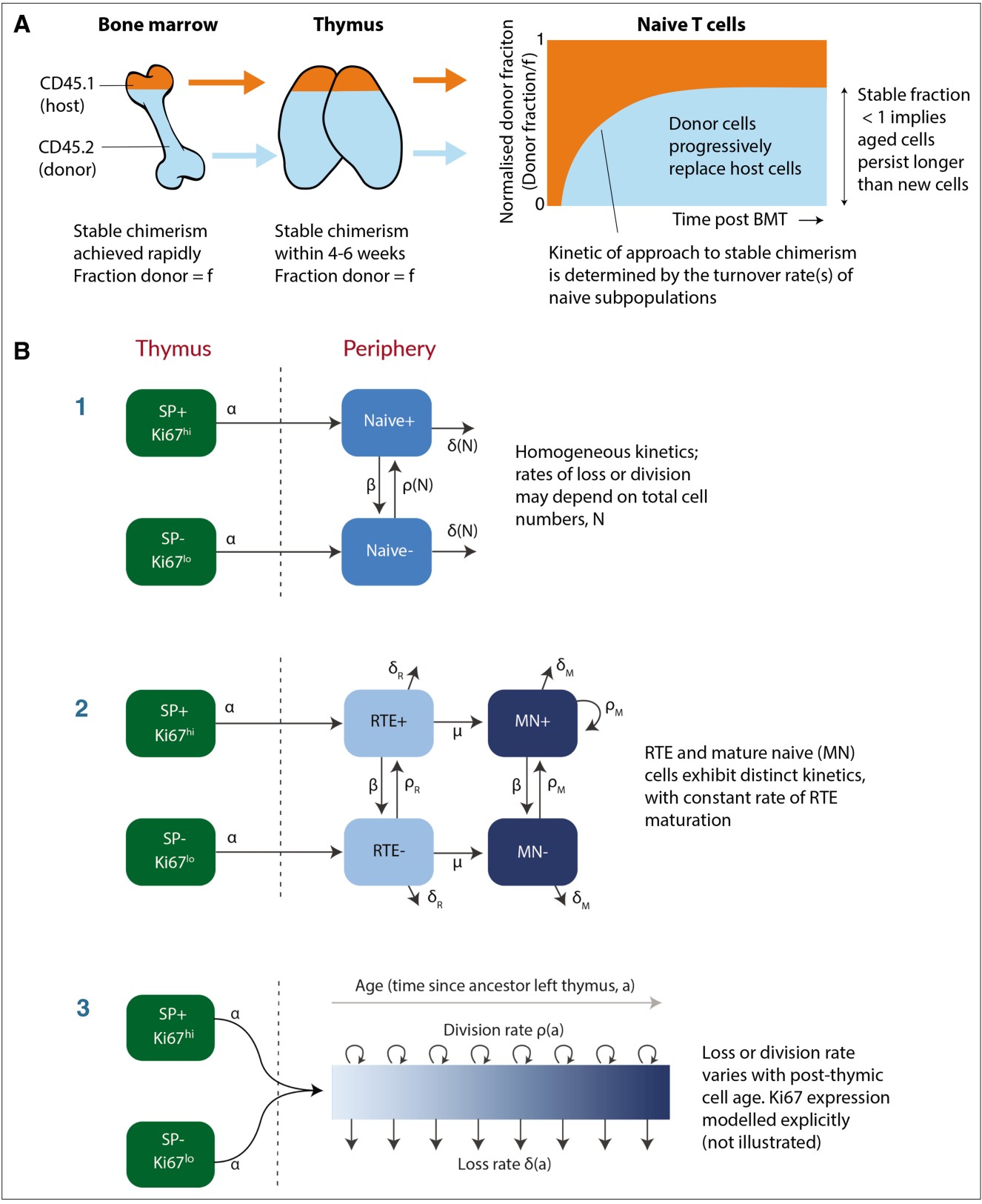

**Figure 1.** Modeling naive T cell dynamics using busulfan chimeric mice. (**A**) Schematic description of the busulfan chimera system, in which congenically labelled donor lymphocytes percolate into peripheral compartments following partial ablation of haematopoietic stem cells and bone marrow transplant (BMT). (**B**) Candidate models of naive T cell dynamics. In all models, we assume Ki67- and Ki67+ cells are exported from the thymus at rates proportional to the numbers of Ki67- and Ki67+ single positive (SP) thymocytes, respectively. We considered three classes of model; (1) Homogeneous,

*Figure 1 continued on next page*

Figure 1 continued

in which all cells are lost at the same rate and divide at the same rate. In the simplest 'neutral' case these rates are constant. We also considered extensions in which loss or division rates were allowed to vary with total cell numbers (density-dependent models). (2) Recent thymic emigrants (RTE) and mature naive (MN) T cells exhibit distinct kinetics, with a constant rate of maturation μ. (3) Loss or division rates vary with post-thymic cell age, $a$. Here we explicitly model the time-evolution of the population density of cells of post-thymic age $a$ with Ki67 expression $k$ at mouse age $t$, $u(a, k, t)$. Mathematical details of all models are given in Appendix 1.

The online version of this article includes the following figure supplement(s) for figure 1:

**Figure supplement 1.** Gating strategies for thymocyte and peripheral naive T cell subsets.

nuclear protein that is expressed following entry into cell cycle and is detectable for approximately 3–4 days afterwards (*Gossel et al., 2017*; *Miller et al., 2018*), and stratified by host and donor cells. We reasoned that this new information would enable us to determine whether cell-age effects are manifest through survival or self-renewal.

To describe these data we explored variants of a structured population model in which either the rate of division or rate of loss of naive T cells varies exponentially with their post-thymic age. These models are three dimensional linear partial differential equations (PDEs) that extend those we described previously (*Hogan et al., 2015*; *Rane et al., 2018*), allowing us to track the joint distribution of cell age and Ki67 expression within the population. A simpler variant of the age-structured model is one that explicitly distinguishes RTE from mature naive T cells, with a constant rate of maturation between two, and allows each to have their own rates of division and loss (*van Hoeven et al., 2017*; *Rane et al., 2018*). We also considered models of homogeneous cell dynamics; the simplest 'neutral' model with uniform and constant rates of division and loss, and density-dependent models that allowed these rates to vary with population size. All models are illustrated schematically in *Figure 1B* and their formulations are detailed in Appendix 1.

Each model was fitted simultaneously to the measured timecourses of total naive CD4 or CD8 T cell numbers, the normalised donor fraction, and the proportions of donor and host cells expressing Ki67. To model influx from the thymus we used empirical functions fitted to the numbers and Ki67 expression levels of late stage single-positive CD4 and CD8 thymocytes (Appendix 2). Assuming that the rate of export of cells from the thymus is proportional to the number of single-positive thymocytes (*Berzins et al., 1998*), we used these functions to represent the rates of production of Ki67$^+$ and Ki67$^-$ RTE with mouse age, up to a multiplicative constant which we estimated. The fitting procedure is outlined in Appendix 3, and detailed in *Verheijen et al., 2020*.

Our analysis confirmed support for the models of cell-age-dependent kinetics (*Figure 2*), with all other candidates, including the RTE model, receiving substantially lower statistical support (*Table 1*; fits shown in *Figure 2—figure supplement 1*). For naive CD4 T cells, we found strongest support for the age-dependent loss model (relative weight = 86%; *Figure 2A*) which revealed that their rate of loss declines as they age, halving roughly every 3 months (*Table 2*). For naive CD8 T cells the age-dependent division model was favoured statistically (relative weight = 85%; *Figure 2B*, dashed lines). However, it yielded extremely low division rates, with recently exported cells having an estimated mean interdivision time of 18 months (95% CI: 14–25), and the division rate increasing only very slowly with cell age (doubling every 10 months). This model was therefore very similar to a neutral, homogeneous model and predicted that the normalised donor fraction approaches 1 in aged mice. This conclusion contradicts findings from our own and others' studies that demonstrated that models assuming homogeneity in naive CD8 T cells failed to capture their dynamics in adult and aged mice (2–20 months old) (*Hogan et al., 2015*; *Rane et al., 2018*; *Reynaldi et al., 2019*).

Any signal of improvement in fitness with cell age, either in loss or division rates, is manifest primarily in an asymptotic value of the normalised donor fraction lower than one. For naive CD8 T cells, the normalised donor fractions at late times post-BMT exhibit considerable scatter (*Figure 2B*, middle row), and so this asymptote is relatively poorly defined. This uncertainty reduces our ability to discriminate between the two age-dependent models based solely on information criteria. For the next phase of analysis, we therefore retained the age-dependent loss model, which had the next highest level of support and was similar by visual inspection (*Figure 2B*, solid lines), as a candidate description of naive CD8 T cell dynamics.

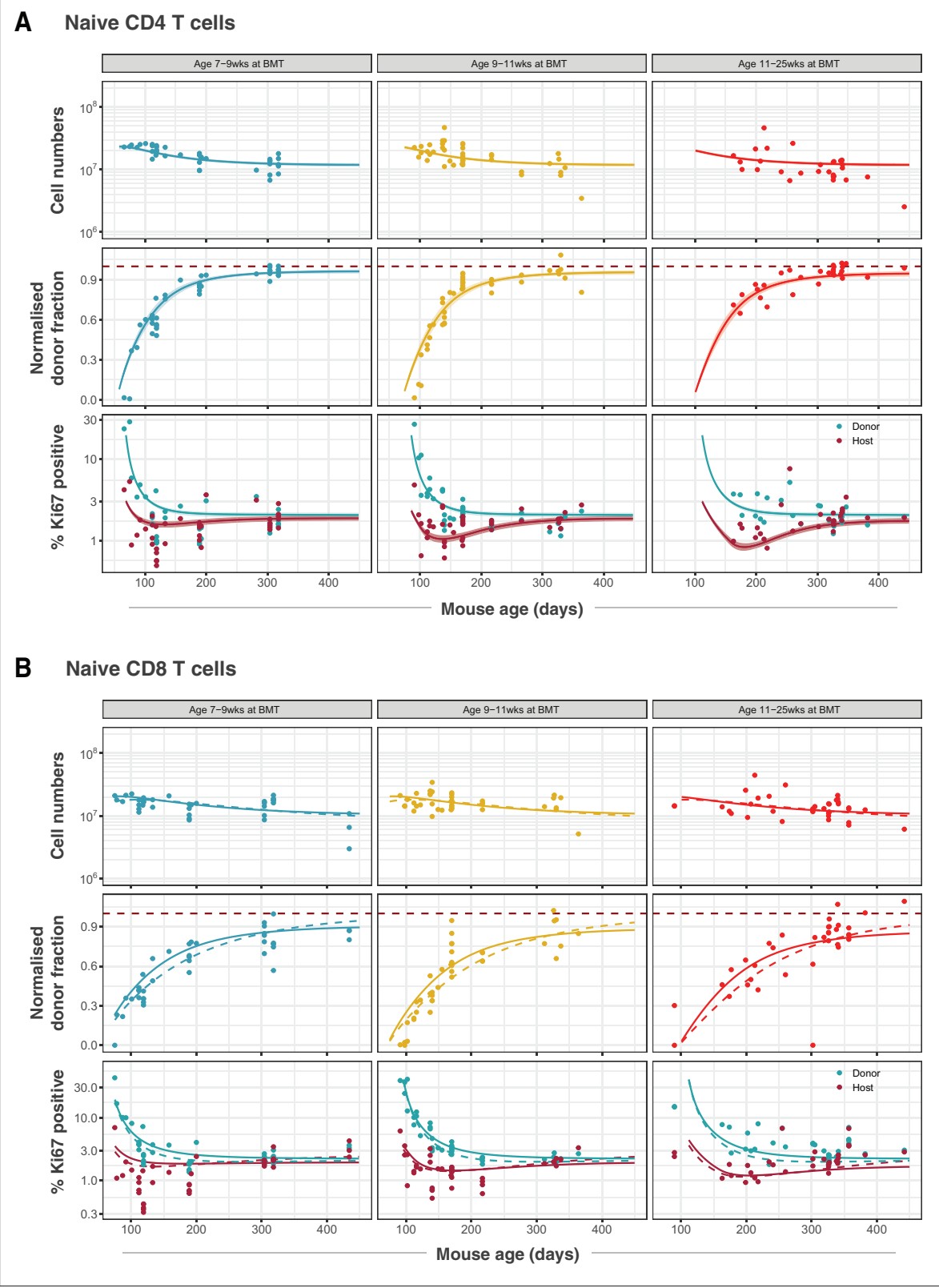

**Figure 2.** Modelling naive CD4 and CD8 T cell dynamics in adult busulfan chimeric mice. (**A**) The best fitting, age-dependent loss model of naive CD4 T cell dynamics describes the timecourses of their total numbers, chimerism and Ki67 expression in mice ($n = 111$) who underwent busulfan treatment and BMT in three different age groups (indicated within grey bars). (**B**) Fits to naive CD8 T cell dynamics ($n = 116$) yielded by the age-dependent division model (dashed lines) and the age-dependent loss model (solid lines). Envelopes indicate the 95% credible interval on the mean of the model

*Figure 2 continued on next page*

*Figure 2 continued*

prediction, generated by sampling from the posterior distributions of model parameters. For clarity, these envelopes are omitted in panel B, to allow visual comparison of the two models.

The online version of this article includes the following figure supplement(s) for figure 2:

**Figure supplement 1.** Fits of alternative models to the data from busulfan chimeric mice.

**Figure supplement 2.** Posterior distributions of key parameters.

## Age-dependent loss models can describe RTE and mature naive CD4 and CD8 T cell kinetics in co-transfer experiments

To challenge these models further, we confronted them with data from a study that compared the ability of RTE and mature naive (MN) CD4 and CD8 T cells to persist following co-transfer to an adult congenic recipient (*Houston et al., 2011*). This study used a reporter mouse strain in which green fluorescent protein (GFP) expression is driven by *Rag2* gene expression elements, and is thus expressed throughout thymic development and for several days following export into the periphery. This is a long-established mouse model in which GFP expression is used as a surrogate marker of RTE status (*Boursalian et al., 2004*). After transferring RTE (GFP+) and MN (GFP-) cells in equal numbers, the RTE:MN ratio within both CD4 and CD8 populations decreased progressively, falling by approximately 50% at 6 weeks (*Figure 3*), indicating that MN T cells persist significantly longer than RTE. We simulated this co-transfer using the models fitted to the data from the busulfan chimeric mice, and found that the age-dependent loss model predicted the trends in the CD4 and CD8 RTE:MN ratios (*Figure 3*, blue lines) while the fitted age-dependent division model, which exhibited very weak age effects, predicted that the ratio would remain close to 1 (*Figure 3*, orange lines). Details of this simulation procedure are given in Appendix 4. These data confirm the presence of strong cell-age effects in naive T cell persistence, and substantially reduce our confidence in the best-fitting model for CD8 T cells, which predicted only a very weak dependence of cell division rates on cell age.

**Table 1.** Ranking of models describing naive CD4 and CD8 T cell dynamics in adult busulfan chimeric mice.

We considered instances of the three classes of model (1–3; illustrated in *Figure 1B*), with each instance fitted simultaneously to the timecourses of total naive T cell numbers, host:donor chimerism, and Ki67 expression within host and donor cells. We indicate the number of fitted quantities; this includes both model parameters and initial conditions. Measures of relative support for each model are expressed as weights, which reflect the average accuracy with which each model predicts out-of-sample data, relative to the other models in consideration. These weights were calculated using the Leave-One-Out cross validation and the Pseudo-Bayesian Model Averaging methods, using the *loo-2.0* package in the *Rstan* library; see Appendix 3 for details.

| Population | Model | Unknowns | Model weight (%) |
|---|---|---|---|
| Naive CD4 | 3 – Loss rate varying with cell age | 4 | 86.3 |
| | 3 – Division rate varying with cell age | 4 | 13.0 |
| | 1 – Neutral | 5 | 0.5 |
| | 2 – RTE and mature naive | 8 | 0.2 |
| | 1 – Density dependent loss | 6 | 0.0 |
| | 1 – Density dependent division (LIP) | 6 | 0.0 |
| Naive CD8 | 3 – Division rate varying with cell age | 4 | 85.0 |
| | 3 – Loss rate varying with cell age | 4 | 9.0 |
| | 1 – Density dependent division (LIP) | 6 | 4.5 |
| | 1 – Density dependent loss | 6 | 1.5 |
| | 2 – RTE and mature naive | 8 | 0.0 |
| | 1 – Neutral | 5 | 0.0 |

**Table 2.** Parameter estimates derived from fitting the age-dependent loss model to data from adult busulfan chimeric mice.

Residence and interdivision times are defined as the inverses of the instantaneous loss rate ($\delta(a)$) and the division rate ($\rho$), respectively. Posterior distributions of model parameters are shown in *Figure 2—figure supplement 2*. CI: credible interval.

| Population | Parameter | Estimate | 95% CI |
|---|---|---|---|
| Naive CD4 | Expected residence time of cells of age 0 (days) | 22 | 18–28 |
| | Time taken for loss rate to halve (days) | 92 | 71–130 |
| | Mean interdivision time (months) | 18 | 16–22 |
| Naive CD8 | Expected residence time of cells of age 0 (days) | 40 | 34–46 |
| | Time taken for loss rate to halve (days) | 146 | 107–206 |
| | Mean interdivision time (months) | 14 | 12–16 |

## Models parameterised using data from adult mice accurately predict the dynamics of naive CD4 T cells in neonates, but not of CD8 T cells

Next, we wanted to characterise the dynamics of naive CD4 and CD8 T cells during the first few weeks of life, and connect the two regimes to build unified models of the dynamics of these populations from birth into old age. Because it takes at least 4 weeks for peripheral donor-derived T cells to be detectable in busulfan chimeras, this system is not suitable for studying cell dynamics in young mice. Instead, we asked whether the models parameterised using data from adult mice could explain dynamics in young mice, and determine what (if any) modifications of the model were needed. We drew on two new data sets. One comprised the numbers and Ki67 expression of naive T cells derived from wild-type mice aged between 5 and 300 days. The other was derived from a cohort of Rag<sup>GFP</sup> reporter mice, in which information about cell age can be gleaned from GFP expression levels. In this strain, intracellular staining for Ki67 is not possible without severely compromising GFP

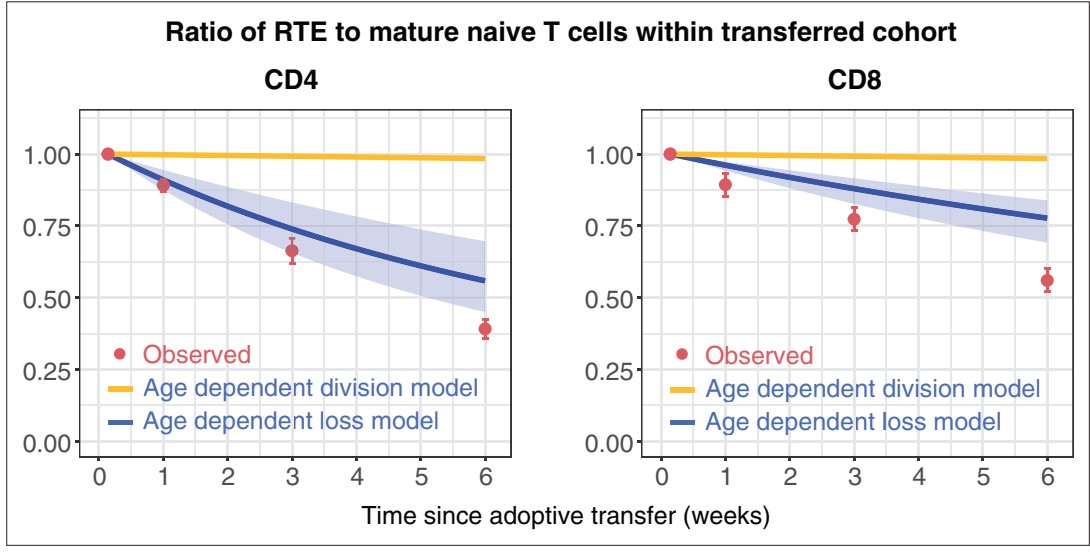

**Figure 3.** Distinct survival kinetics of RTE and mature naive T cells favour models with strong cell-age effects. We simulated the co-transfer experiment described by *Houston et al., 2011* in which RTE from 5- to 9-week-old Rag<sup>GFP</sup> reporter mice were co-transferred with equal numbers of mature naive (MN) T cells from mice aged 14 weeks or greater to congenic recipients. Red points represent their observed RTE:MN ratios. We then used the models fitted to the data from busulfan chimeric mice (*Figure 2*) to predict the outcome of this co-transfer experiment, with the age-dependent division model shown in orange, and the age-dependent loss model in blue. The pale blue envelopes show the median and 2.5% and 97.5% quantiles of the RTE:MN ratio predicted by the models, obtained by sampling from the posterior distribution of parameters. This envelope was too narrow to be shown for the age-dependent division models (orange lines).

fluorescence. Therefore, we also introduced a Ki67^RFP reporter construct (*Basak et al., 2014*) to the strain to generate Rag^GFPKi67^RFP dual reporter mice. Tracking GFP and RFP expression simultaneously allows us to study the kinetics and division rates of RTE, which are enriched for GFP^+ cells, and of mature naive T cells, which are expected to have largely lost GFP. We could then directly confront the models derived from adult mice with these new data.

*Figure 4A and B* show the numbers of naive CD4 and CD8 T cells and their Ki67 expression frequencies in three cohorts of mice – Rag^GFPKi67^RFP dual reporter mice aged between 10 and 120 days, wild-type mice, and adult busulfan chimeras in which host and donor cells were pooled. The red curves show the predictions of the cell-age-dependent loss models, which were fitted to the busulfan chimera data (red points) and extrapolated back to 1 day after birth. The dual reporter mice also yielded measurements of the co-expression of GFP and Ki67. To predict the kinetics of GFP^+Ki67^− and GFP^+Ki67^+ proportions (*Figure 4C and D*), we needed to estimate only one additional parameter – the average duration of GFP expression. We assume that RTE become GFP-negative with first order kinetics at a rate defined both by the intrinsic rate of decay of GFP and the threshold of expression used to define GFP^+ cells by flow cytometry. Our estimates of the mean duration of GFP expression within CD4 and CD8 RTE were similar (11 and 8 days, respectively). Details of how we connected GFP measurements to the age-structured models are provided in Appendix 5, and a description of the process of predicting neonatal T cell dynamics is given in Appendix 6.

Strikingly, the model of naive CD4 T cell dynamics in adult chimeric mice captured the total numbers and Ki67 expression of these cells in neonates remarkably well (*Figure 4A*), as well as the dynamics of Ki67^- and Ki67^+ RTE as defined by GFP expression (*Figure 4C*). This agreement indicates that the high level of Ki67 expression in naive CD4 T cells early in life does not reflect increased rates of division or LIP, but is rather inherited from precursors within the neonatal thymus, a large fraction of which undergo cell division (Appendix 2).

For naive CD8 T cells the cell-age-dependent loss model accurately predicted cell dynamics in both the reporter and wild-type mice back to approximately 3 weeks of age, but underestimated Ki67^+ frequencies in neonatal mice (*Figure 4B*, right panel), suggesting that naive CD8 T cells exhibit distinct dynamics very early in life. Intuitively, this mismatch can be explained in two ways: either CD8 RTE are lost at a higher rate in neonates than in adults or they divide more rapidly. In the former, a greater proportion of GFP^+ Ki67^+ RTE will be lost before they become Ki67^- and so the predicted proportion of cells that are GFP^+ Ki67^- will be lower (*Figure 4D*, centre panel). In the latter, the GFP^+ Ki67^+ proportion will increase (*Figure 4D*, right panel). Therefore, to explain naive CD8 T cell dynamics in neonates the basic model of cell-age-dependent loss in adults can be extended in two ways, modulating either the division or loss rate early in life.

## Naive CD8 T cells are lost at a higher rate in neonates than in adults

To distinguish between these possibilities we turned to a study by *Reynaldi et al., 2019*, who used an elegant tamoxifen-driven CD4-Cre^ERT2-RFP reporter mouse model to track cohorts of CD8 T cells released from the thymus into the peripheral circulation of animals of varying ages. In this model, a pulse of tamoxifen permanently induces RFP in cells expressing CD4, including CD4^+CD8^+ double-positive thymocytes. The cohort of naive CD8 T cells deriving from these precursors continues to express RFP in the periphery and timecourses of their numbers in individual mice were estimated with serial sampling of blood. These timecourses showed that the net loss rate of naive CD8 T cells appears to slow with their post-thymic age, and the rate of loss of cells immediately following release from the thymus appears to be greater in neonates than in adults (*Figure 5A*). Without measures of proliferation, these survival curves reflect only the net effect of survival and self-renewal. Nevertheless, we reasoned that confronting our models with these additional data, and triangulating with inferences from other datasets, would allow us to identify a 'universal' model of naive CD8 T cell loss and division across the mouse lifespan.

We re-analysed the data from Reynaldi et al. using a Bayesian hierarchical approach (Appendix 7) to explain the variation in the kinetics of loss of these cohorts of cells across animals and age groups. Since there was no readout of cell division in this system, we simplified the cell-age-dependent loss model by combining division and loss into a net loss rate $\lambda(a)$. We then fitted this model to the time-courses of labelled naive CD8 T cells across the different treatment groups. We tested four possibilities in which either the initial numbers of labelled cells ($N_0$) and/or the net loss rate of cells of age 0

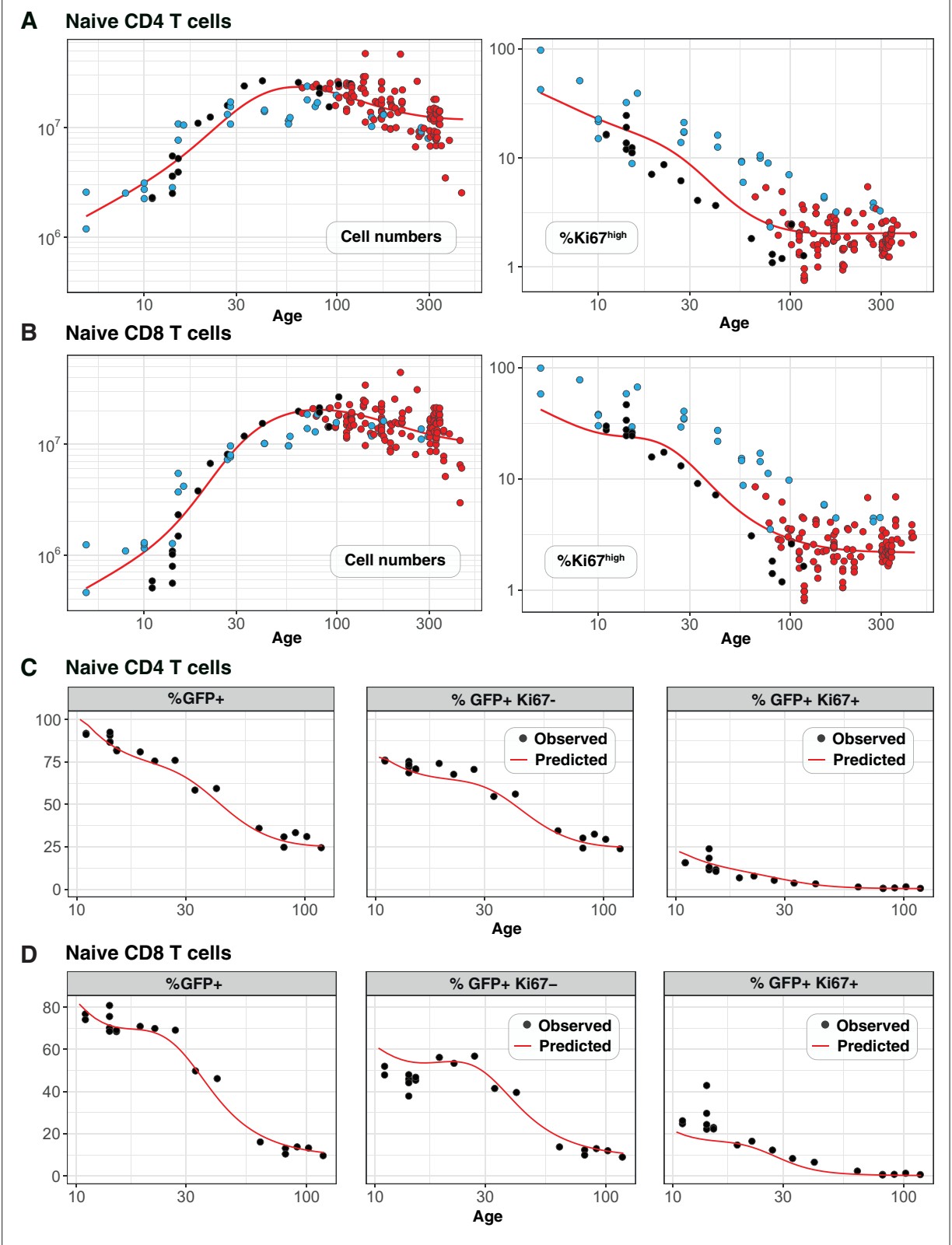

**Figure 4.** Predicting the kinetics of establishment of naive CD4 and CD8 T cell pools in early life. Panels A and B: For naive CD4 and CD8 cells, we extrapolated the age-dependent loss models (red curves) that were fitted to data from adult busulfan chimeric mice (red points) back to age 1 day. We compared these predicted trajectories with independent observations of naive T cell numbers and Ki67 expression in wild-type mice aged between 5–300 days ($n = 34$ mice, blue points), and from Rag$^{GFP}$Ki67$^{RFP}$ reporter mice ($n = 19$ mice, black points). Panels C and D: We then estimated one

*Figure 4 continued on next page*

*Figure 4 continued*

additional parameter – the expected duration of GFP expression – by fitting the age-dependent loss model to the timecourses of total numbers of naive CD4 and CD8 GFP⁺ cells in these reporter mice (leftmost panels). We could then predict the timecourses of the percentages of GFP⁺Ki67⁺ and GFP⁺Ki67⁻ cells (centre and right panels).

($\lambda_0$) varied across groups or animals as normally-distributed hyper-parameters. The model in which $N_0$ was specific to each mouse and $\lambda_0$ was specific to each age group gained 100% relative support (*Appendix 7—table 1*; fits in *Figure 5A*). This model confirmed that CD8 RTE are indeed lost at a significantly higher rate in the younger groups of mice (*Figure 5B*). We then described this decline in $\lambda_0$ with mouse age empirically with a sigmoid (Hill) function, $\lambda_0(t)$ (*Figure 5B*, solid line) and used it to replace the discrete group-level variation in $\lambda_0$ within the hierarchical age-structured model (Appendix 7). This 'universal' model, in which the loss rate of naive CD8 T cells declines with cell age but begins at higher baseline levels early in life, explained the data from Reynaldi et al. equally well, visually and statistically (difference in the expected log pointwise predictive density, elpd_loo=3.4; differences

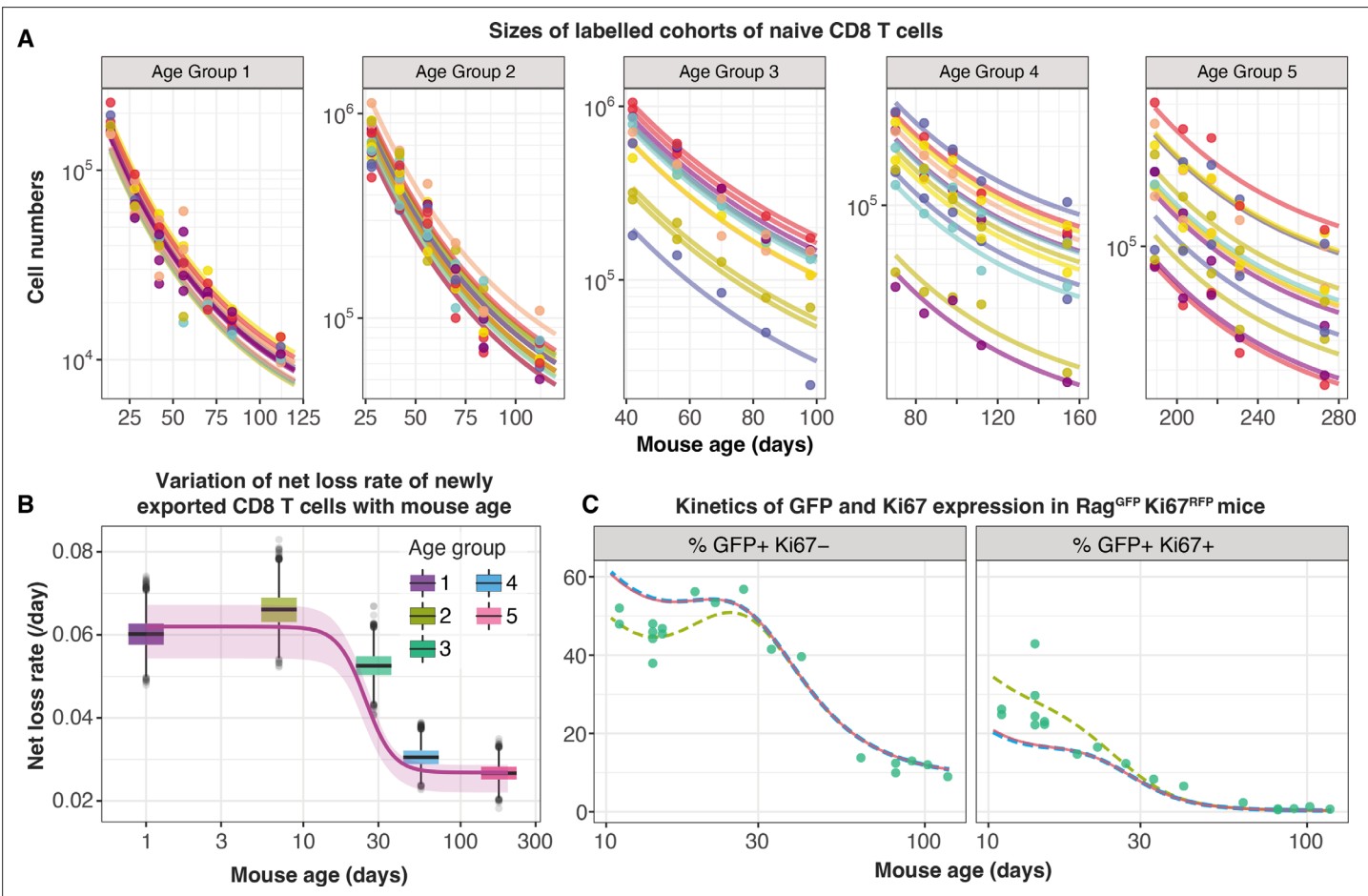

**Figure 5.** Tracking the persistence of cohorts of naive CD8 T cells *in vivo* – an analysis of data from *Reynaldi et al., 2019*. (**A**) Fitting the age-dependent loss model to the estimated numbers of time-stamped naive CD8 T cells in CD4-Cre^ERT2 reporter mice ($n = 66$) treated with tamoxifen at different ages and sampled longitudinally. We used a hierarchical modelling framework and show mouse-specific fits to these timecourses (colours indicate different animals, dots are observations and lines are model fits). In the best fitting model, estimates of initial cell numbers were mouse-specific, while the net loss rate of RTE of age 0 ($\lambda_0 = \lambda(a = 0)$) was specific to each mouse age group. (**B**) Corresponding estimates of $\lambda_0$ for each age group of mice (black horizontal bars), with mouse-specific estimates (grey points) and the fitted, empirical description of $\lambda_0$ with mouse age (see Appendix 7, *Equation 42*). (**C**) Predicting the kinetics of the percentages of GFP⁺ Ki67⁻ and GFP⁺ Ki67⁺ CD8 T cells using the age-dependent loss model, including neonatal age effects in either the loss rate (green dashed line) or in the division rate (blue dashed line). The red line (partly concealed by the blue dashed line) shows the predictions of the original model fitted to the adult busulfan chimeric mice, with no mouse age effects.

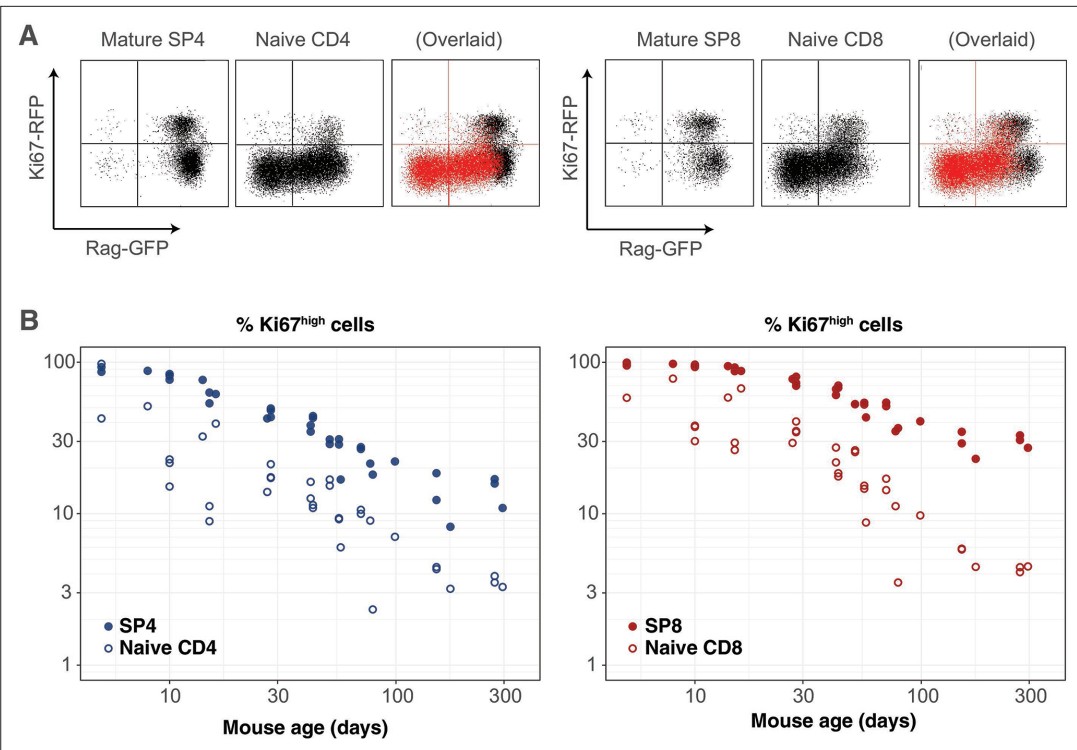

**Figure 6.** Markers of proliferation among naive T cells derived from very recent thymic emigrants. (**A**) Flow cytometry analyses of late stage single positive thymocytes and naive CD4 and CD8 T cells from lymph nodes in a 41-day-old Rag$^{GFP}$ Ki67$^{RFP}$ reporter mouse, showing that Ki67 expression among naive T cells is largely restricted to GFP$^+$ RTE. In the 'overlaid' panels, naive T cells are shown in red and mature SP thymocytes in black. (**B**) Data from a cohort of wild-type mice showing that Ki67 levels in SP thymocytes and peripheral naive T cells correlate throughout life (Spearman's rank correlation coefficient; $\rho$ = 0.90 (CD4), 0.94 (CD8); both $p<10^{-15}$).

<4 typically indicate that two models have similar predictive performance (*Sivula et al., 2020*). See Appendix 3 for details of the calculation of elpd$_{loo}$ values.)

This analysis shows that the baseline net loss rate of CD8 RTE declines from the age of ~3 weeks and stabilises at a level approximately 50% lower by age 9 weeks (*Figure 5B*). Therefore, newly exported naive CD8 T cells are lost at a higher rate in neonates than in adults, or they divide more slowly. Only the former is consistent with our inference from the Rag/Ki67 dual reporter mice. Indeed, we confirmed that simulating the age-dependent loss model from birth with a lower baseline division rate in neonates than in adults failed to improve the description of the early trajectories of the frequencies of GFP$^+$ Ki67$^-$ and GFP$^+$ Ki67$^+$ naive CD8 T cells (*Figure 5C*, blue dashed line). In contrast, increasing the baseline loss rate in neonates according to the function we derived from the data in Reynaldi et al. (Appendix 7) captured these dynamics well (*Figure 5C*, green dashed line).

In summary, we find that naive CD8 T cells rarely divide, increase their capacity to survive with cell age, and those generated within the first few weeks of life are lost at a higher baseline rate than those in adults.

## Ki67 expression within naive CD4 and CD8 T cells in adult mice is almost entirely a residual signal of intra-thymic proliferation

Our analyses are consistent with earlier reports that naive T cells in mice divide very rarely (*Modigliani et al., 1994*; *den Braber et al., 2012*; *Hogan et al., 2015*). By explicitly modeling the kinetics of quiescent and recently divided cells, we can also explain the apparently contradictory observation that more than 60% of naive CD4 and CD8 T cells express Ki67 early in life, declining to 2–3% by 3 months of age (*Figure 4*). We argue that this pattern, rather than being an indication of lymphopenia-induced proliferation early in life fading to low-level but appreciable self-renewal in adults, is instead just a shadow of intrathymic division; Ki67 among peripheral naive T cells is almost entirely derived from

cells that divided in the thymus and were exported within the previous few days. This conclusion emerged from the modelling of the busulfan chimera data but is also directly evident from the Rag$^{GFP}$ Ki67$^{RFP}$ reporter mice, in which Ki67-RFP expression among naive T cells was exclusively found on GFP$^{high}$ peripheral RTE, and was a continuum of the expression by mature single-positive (SP) thymocytes (*Figure 6A*). This inheritance of expression from the thymus is also reflected in the high degree of correlation between the frequencies of Ki67$^+$ cells among SP thymocytes and peripheral naive T cells throughout life, observed in wild-type mice (*Figure 6B*).

This result also gives an intuitive explanation of the trajectories of Ki67 expression within donor and host cells in the busulfan chimeric mice, which are distinct soon after BMT but converge after 6–12 months (*Figure 2*). This behaviour does not derive from any intrinsic differences between host and donor T cells, but rather from the distinct age profiles of the two populations. Following BMT, the rate of production of host naive T cells declines substantially, as the procedure typically results in 80%–90% replacement of host HSC with donor HSC. Since Ki67 is seen almost exclusively within very recent thymic emigrants, the frequency of Ki67-expressing host naive T cells then declines rapidly. Conversely, new donor-derived naive T cells are initially highly enriched for Ki67$^+$ cells. The frequencies of Ki67$^+$ cells within the two populations then gradually converge to pre-transplant levels as aged Ki67$^-$ donor cells accumulate, and host-derived naive T cells equilibrate at lower numbers.

## Discussion

Our previous analyses suggested naive T cells operate autonomously and compensate for the gradual decline in thymic output with age by increasing their ability to persist with time since they leave the thymus in both adult mice (*Rane et al., 2018*) and in humans (*Mold et al., 2019*). Here, we show through the modelling of a range of datasets that naive T cell adaptation in mice manifests primarily through a progressive decrease in their loss rate, and that they divide very rarely if at all, with mean interdivision times of at least 14 months. This means that throughout the mouse lifetime, newly made CD4 and CD8 RTE are lost at faster rates than their mature counterparts, predicting the preferential retention and accumulation of clones exported early in life. The lack of peripheral expansion combined with high levels of thymic export throughout life implies that, in mice, the majority of the naive T cell repertoire is made up of small and long-lived TCR clones. This interpretation is consistent with studies showing enormous diversity within naive TCR repertoires in mice (*Gonçalves et al., 2017*) and supports the idea that any hierarchy within it is shaped by the generational frequencies of individual clones in the thymus, rather than by peripheral expansions (*Quigley et al., 2010*). However, we observed a remarkably high degree of intrathymic proliferation in young mice, with close to 100% of late-stage CD62L$^{hi}$ SP thymocytes expressing Ki67 in neonates, declining to approximately 20% over the first 3 months of life (*Figure 6B* and Appendix 2). Ki67 expression within these mature SP populations derives exclusively from cell division after TCR rearrangement and positive selection are complete. This observation implies that naive T clones generated in neonatal mice, which will ultimately be over-represented in older mice, may be substantially larger on average than those exported from adult thymi.

It is well-established that proliferative self-renewal plays a much more important role in naive T cell dynamics in humans than in mice (*den Braber et al., 2012*), which may compensate in part for the quite severe atrophy of the thymus that progresses from young adulthood onwards (*Steinmann et al., 1985*). However, we and others have shown that, similar to mice, the net loss rates of naive T cells in humans appear to fall with cell age (*Johnson et al., 2012*; *Mold et al., 2019*). Thus, progressive, cell-intrinsic increases in the homeostatic fitness of naive T cells may be a common mechanism. Whether this adaptation in humans occurs through changes in the capacity to survive or to self-renew, or both, is unclear. In any case the implication is that, as in mice, naive T cell age distributions in humans become disproportionally weighted toward older cells, or clones, over time. The combination of cell proliferation and extended lifespans, which give more time for cell fitness disparities to widen, may underlie the even broader distributions of naive T cell clone sizes observed in humans (*Qi et al., 2014*; *Mora and Walczak, 2019*; *de Greef et al., 2020*).

Our fitted age-dependent loss models showed agreement with the trends demonstrated by *Houston et al., 2011*, in which mature naive (MN) CD4 and CD8 T cell persist longer than RTE, but underestimated the extent of enrichment of MN cells (*Figure 3A*). This mismatch may derive in part from uncertainties in the age-distributions of the transferred T cells in their experiments, and from our

need to specify a cut-off in cell age associated with the definition of RTE as GFP-positive. It is also possible that the cell manipulations involved in adoptive transfer had a differential impact on RTE and MN cell survival. The essential point here, however, is that the clear disparity of kinetics of the two transferred populations weighs against models exhibiting weak cell-age effects.

Another modelling study also demonstrated that CD4 RTE are lost more rapidly than MN CD4 T cells (*van Hoeven et al., 2017*). They estimated that the loss rate of CD4 RTE is 0.063 day$^{-1}$, translating to a residence time of 15 days (95% CI: 9–26), and is roughly four times shorter than the 66 day (52–90) residence time of MN CD4 T cells. Our results agree closely. We estimate that CD4 RTE (cells of age 0) have an expected residence time of 22 (18–28) days, doubling approximately every 3 months, such that in a 12-week-old mouse, the mean residence time of MN CD4 T cells aged 21 days or greater is roughly 60 days. In contrast, van Hoeven et al. concluded that naive CD8 T cells are a kinetically homogeneous population with a mean residence time of 76 (42–83) days. With our favoured age-dependent loss model, we estimate that CD8 RTE initially have an expected residence time of 40 (18–28) days, doubling every ~5 months. However, our predicted average residence time of MN CD8 T cells (aged >21 days) in a 12 week old mouse was approximately 76 days, which agrees with their estimate. We included a similar RTE/MN model in our analysis (*Figure 1B*) and found that for CD8 T cells it received statistical support comparable to a neutral model of constant division and loss, in line with their analysis. Therefore, our different conclusions may stem in part from the specification of our models. It would be instructive to analyse the data from their thymic transplantation and heavy water labelling studies with the age-structured models we consider here. Another puzzle is that our result and those of *van Hoeven et al., 2017*, *Houston et al., 2011*, and *Tsukamoto et al., 2009* are all at odds with the study of *Dong et al., 2013* who observed that CD4 GFP$^+$ RTE from Rag-GFP reporter mice persisted better than bulk naive CD4 T cells from age-matched donor mice, one week after co-transfer. We are unable to explain this observation, although we speculate that differential survival may have been influenced by the manipulation step of labelling the bulk naive T cell cohort, but not the RTE, with a fluorescent dye (CFSE).

The pioneering studies by Berzins et al. showed substantial and proportional increases in T cell numbers in mice transplanted with 2, 6 and 9 thymic lobes *Berzins et al., 1998*; *Berzins et al., 1999*. They concluded that this increase corresponds to the accumulation of RTE exported in the previous 3 weeks. In absence of any homeostatic regulation, the increase in the sizes of the naive CD4 and CD8 T cell pools under hyperthymic conditions is determined by the change in thymic output and by RTE lifespans. Our estimates of these lifespans (roughly 22 and 40 days for CD4 and CD8, respectively) are in line with their estimate of 3 weeks (*Berzins et al., 1999*). Indeed, simulating the transplantation of 6 thymic lobes using the age-dependent loss models and parameters derived from busulfan chimeric mice recapitulates their observations (Appendix 7).

Reynaldi et al. used a novel fate-mapping system to demonstrate that the net loss rate of naive CD8 T cells (loss minus self-renewal) declines with their post-thymic age and is higher for CD8 RTE in neonates than in adults *Reynaldi et al., 2019*. Our addition to this narrative was to reanalyse their data with a more mechanistic modeling approach to isolate the effects of division and loss, and to calculate a functional form for the dependence of the CD8 RTE loss rate on mouse age. In conjunction with our analysis of data from Rag/Ki67 dual reporter mice we inferred that the baseline loss rate of naive CD8 T cells immediately following release from the thymus is higher in neonates than in adults, while the rate of division is close to zero throughout the mouse lifespan, and independent of host and cell age. The higher loss rate of CD8 RTE in neonates may derive from high rates of differentiation into memory phenotype cells rather than impaired survival. This idea is consistent with the rapid accumulation of virtual memory CD8 T cells in the periphery during the postnatal period (*Akue et al., 2012*, *Smith et al., 2018*). However, we found no evidence for a similar process among naive CD4 T cells. We recently showed that increasing the exposure to environmental antigens boosts the generation of memory CD4 T cell subsets early in life, but not in adulthood (*Hogan et al., 2019*). It may be that in the young specific pathogen-free mice we studied here, any such elevated flux out of the naive CD4 T cell pool due to activation, which occurs before clonal expansion, was too low for our analysis to detect. We speculate that any dependence of naive T cell residence times on host age may be even more pronounced in truly wild mice and in humans, given their extensive exposure to environmental and commensal antigens immediately following birth.

We do not explicitly model the mechanisms underlying adaptation in cell persistence. Modulation of sensitivity to IL-7 and signaling via Bcl-2 associated molecules has been implicated in increasing naive T cell longevity (*Tsukamoto et al., 2010*; *Houston et al., 2011*) and is consistent with the outcome of co-transfer experiments. It is also possible that increased persistence derives additionally from a progressive or selective decrease in naive T cells' ability to be triggered into effector or memory subsets. Studying the dynamics of naive T cells in busulfan chimeras generated using bone marrow from TCR-transgenic donor mice may help us untangle the contributions of survival and differentiation to the increase in their residence time with their age.

An alternative to the adaptation model is one of selection, in which each cell's survival capacity (loss rate) is determined during thymic development, drawn from a distribution, and subsequently fixed for its lifespan (*Dowling et al., 2005*; *Rane et al., 2018*). As an individual ages, naive T cells with intrinsically longer expected residence times will then be selected for. Indirect support for such a mechanism comes from the observation that low-level TCR signalling is essential for naive T cell survival (*Seddon and Zamoyska, 2002*; *Martin et al., 2006*), suggesting that the ability to gain trophic signals from self-peptide MHC ligands may vary from clone to clone. We have also shown that different TCR-transgenic naive T cell clones have different capacities for proliferation in lymphopenic hosts (*Hogan et al., 2013*). However, one prediction of a selective model based on heterogeneity in TCR affinity alone is that TCR transgenic T cells co-transferred from young and old hosts would be lost at identical rates. One such experiment still saw that older cells exhibited a fitness advantage over younger ones (*Tsukamoto et al., 2009*). Therefore, while we cannot rule out a pre-programmed (and possibly TCR-specific) element to each naive T cell's life expectancy, it is clear that they undergo progressive changes in their fitness, expressed in the adaptation models we have considered here.

Naive T cells proliferate under severely lymphopenic conditions in mice (*Almeida et al., 2001*; *Yates et al., 2008*; *Hogan et al., 2013*), a phenomenon that has contributed to the idea that quorum-sensing (through resource competition, for example) may act to regulate naive T cell numbers. However, lymphopenia-induced proliferation is associated with the acquisition of a memory-like phenotype (*Cho et al., 2000*; *Min et al., 2003*; *Min and Paul, 2005*; *Hogan et al., 2013*). The observation that this process occurs in healthy neonatal mice was taken to indicate that they are lymphopenic to some degree (*Min et al., 2003*), but it has been shown since that there are constitutive flows from the naive CD4 and CD8 to memory-phenotype T cell pools throughout life under replete conditions (*van Hoeven et al., 2017*; *Gossel et al., 2017*; *Hogan et al., 2019*), and that this transition can occur soon after release from the thymus (*van Hoeven et al., 2017*). It is therefore not clear that young mice are functionally lymphopenic, nor that any compensatory processes support the production or maintenance of truly naive T cells early in life. In line with this, our analyses of neonatal mice revealed no evidence of increased rates of self-renewal, nor any reduction of cell loss rates, that would act to boost or preserve naive T cell numbers in the early weeks of life; neither did we need to invoke feedback regulation of their kinetics in adulthood. We showed previously that apparent density-dependent effects on naive T cell survival following thymectomy can also be explained by adaptive or selective processes (*den Braber et al., 2012*; *Rane et al., 2018*). Similarly, in humans, any regulation of a natural set-point appears to be incomplete at best; naive T cell numbers in HIV-infected adults typically do not normalise following antiretroviral therapy (*Hazenberg et al., 2000*), and recovery from autologous haematopoietic stem cell transplant results in persistent perturbations of T cell dynamics (*Baliu-Piqué et al., 2021*). *Dutilh and de Boer, 2003* showed that explaining the kinetics of decline in TREC frequencies in human naive T cells requires an increase in either cell division or survival with age, as naive T cell numbers decline. They ascribed this to a density-dependent, homeostatic mechanism, but again cell-intrinsic adaptation or selection could underlie the phenomenon. Therefore, the idea of naive T cell homeostasis over the life course, in the sense of compensatory or quorum sensing behaviour, may well be largely a theoretical concept. Selection pressures that shaped the evolution of lymphocyte development are most likely to have been exerted on the establishment of T cell compartments and immunity that would support host survival to reproductive age, and would have little traction upon T cell behaviour into old age. Perhaps a better model, in both mice and humans, is the traditional understanding in which the thymus drives the generation of the bulk of the naive T cell pool in the early life, and thereafter naive T cell repertoires coast out into old age in a cell-autonomous manner.

## Materials and methods

### Generating busulfan chimeric mice

SJL.C57Bl/6J (CD45.1.B6) mice were treated with optimised low doses of busulfan to deplete HSC but leave peripheral T cell subsets intact. HSC were reconstituted with congenically-distinct, T cell depleted bone marrow from C57Bl/6J donors to generate stable chimeras. Details of the protocols are given in *Hogan et al., 2017*.

### Mice

Mki67tm1.1Cle/J (Ki67-RFP) mice were generously provided by the laboratory of Prof. Hans Clevers (Hubrecht Institute, KNAW and University Medical Centre Utrecht, Utrecht, The Netherlands; *Basak et al., 2014*). FVB-Tg(Rag2-EGFP) 1Mnz/J mice were from Jax Laboratories (strain 005688). Ki67-RFP x Rag2-EGFP F1 mice were subsequently backcrossed to a C57Bl/6J background for seven generations. Busulfan chimeric mice and wild-type control mice were housed in conventional animal facilities at the UCL Royal Free Campus, London, UK (UCL). Mice were housed in individually ventilated cages and drank irradiated water. All the animals were handled according to UK home office regulations (licence PPL PP2330953) and institutional animal care and use committee (IACUC) protocols at University College London.

### Flow cytometry

Single-cell suspensions were prepared from the thymus, spleen and lymph nodes of busulfan chimeric mice, wildtype control mice, or germ-free mice. Cells were stained with the following monoclonal antibodies and cell dyes: CD45.1 FITC, CD45.2 FITC, CD45.2 AlexaFluor700, TCR-$\beta$ APC, $CD4^+$ PerCP-eFluor710, CD44 APC-eFluor780, CD25 PE, CD25 eFluor450, CD25 PE-Cy7, CD62L eFluor450, NK1.1 PE-Cy7 (all eBioscience), CD45.1 BV650, CD45.2 PE-Dazzle, TCR-$\beta$ PerCP-Cy5.5 $CD4^+$ BV711, CD44 BV785, CD25 BV650 (all Biolegend), CD62L BUV737 (BD Biosciences), LIVE/DEAD near-IR and LIVE/DEAD blue viability dyes. For Ki67 staining, cells were fixed using the eBioscience Foxp3 /Transcription Factor Staining Buffer Set and stained with either anti-mouse Ki67 FITC or PE (both eBioscience). Cells were acquired on a BD LSR-Fortessa flow cytometer and analysed with Flowjo software (Treestar). See *Figure 1—figure supplement 1* for the gating strategy used to identify mature single positive thymocytes and peripheral naive subsets, and gates to measure Ki67 frequencies.

### Mathematical modelling and statistical analysis

We fitted a set of candidate mathematical models (described in Appendix 1) to the data from adult busulfan chimeric mice, using empirical descriptions of the pool sizes and $Ki67^+$ fraction within SP thymocytes to define thymic influx (Appendix 2). Specifically, we fitted simultaneously to the time courses of total cell counts, normalised donor fraction and the fraction of cells that were $Ki67^+$ within donor and host subsets of naive CD4 and CD8 T cells. We used a Bayesian estimation approach using *R* and *Stan*. Code and data used to perform model fitting, and details of the prior distributions for parameters, are available at this linked Github repository. Models were ranked based on information criteria estimated using the Leave-One-Out (LOO) cross validation method (*Vehtari et al., 2015*; *Vehtari et al., 2016*), described in Appendix 3. Appendix 4 describes how we simulated the co-transfer experiment performed by *Houston et al., 2011*, using the age-structured PDE model (Appendix 1) with parameters estimated from fits to the busulfan chimeric mouse data. To predict the dynamics of naive T cells in neonatal mice, we constructed a mapping between cell age and GFP expression to predict the kinetics of $GFP^+$ and $Ki67^+$ cells in $Rag^{GFP}$ $Ki67^{RFP}$ reporter mice aged between 11 days and 4 months (Appendix 5), and the models fitted to data from adult mice were extrapolated back to near birth (Appendix 6). To re-analyze longitudinal data from *Reynaldi et al., 2019*, tracking the survival of cohorts of naive CD8 T cells within different age groups of mice, we used a hierarchical Bayesian modelling approach (Appendix 7).

## Additional information

### Funding

| Funder | Grant reference number | Author |
| --- | --- | --- |
| National Institutes of Health | R01AI093870 | Andrew J Yates |
| University College London | Medical Research Council UK programme grant MR/P011225/1 | Benedict Seddon |

The funders had no role in study design, data collection and interpretation, or the decision to submit the work for publication.

### Author contributions

Sanket Rane, Formal analysis, Investigation, Methodology, Software, Visualization, Writing – original draft, Writing – review and editing; Thea Hogan, Data curation, Investigation, Methodology, Visualization, Writing – review and editing; Edward Lee, Formal analysis, Investigation, Software, Visualization, Writing – review and editing; Benedict Seddon, Conceptualization, Funding acquisition, Investigation, Methodology, Project administration, Resources, Supervision, Writing – original draft, Writing – review and editing; Andrew J Yates, Conceptualization, Formal analysis, Funding acquisition, Investigation, Methodology, Project administration, Resources, Supervision, Writing – original draft, Writing – review and editing

### Author ORCIDs

Benedict Seddon (iD) http://orcid.org/0000-0003-4352-3373
Andrew J Yates (iD) http://orcid.org/0000-0003-4606-4483

### Ethics

All of the animals were handled according to UK home office regulations (licence PPL PP2330953) and institutional animal care and use committee (IACUC) protocols at University College London.

### Decision letter and Author response

Decision letter https://doi.org/10.7554/eLife.78168.sa1
Author response https://doi.org/10.7554/eLife.78168.sa2

### Data availability

All code and data used in this study are available at https://github.com/sanketrane/T_cell_dynamics_birth-death (copy archived at swh:1:rev:6e17ad8936bb34e966b5b920a943b6355981124e).

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

## Appendix 1

### Models of naive T cell maintenance

#### Neutral model

We assume that the naive T cell pool is a kinetically homogeneous population that self-renews through a constant rate of division $\rho$, and is lost by death and differentiation at a constant rate $\delta$. The inverse of $\rho$ is the mean interdivision time, and the inverse of $\delta$ is the mean residence time of a cell. Thymic influx is the product of the *per capita* rate of influx $\alpha$ and the timecourse of the size of SP population $S(t)$, which is described empirically (see Appendix 2). We model the dynamics of Ki67$^+$ ($N^+$) and Ki67$^-$ ($N^-$) cells using the following ODE model;

$$\frac{dN^-}{dt} = \alpha S(t)(1 - \epsilon(t)) + \beta N^+(t) - (\rho + \delta) N^-(t), \tag{1}$$

$$\frac{dN^+}{dt} = \alpha S(t)\epsilon(t) + \rho (2 N^-(t) + N^+(t)) - (\beta + \delta) N^+(t). \tag{2}$$

Here, $\beta$ is the rate of loss of Ki67 expression after mitosis, and $\epsilon$ is the Ki67$^+$ fraction among cells immediately after export from the thymus. We assumed *Equation 2* hold identically for host and donor cells and solved them to derive the solutions to the following:

$$\text{Total cell numbers} = N(t) = N^+_{\text{donor}}(t) + N^+_{\text{host}}(t) + N^-_{\text{donor}}(t) + N^-_{\text{host}}(t),$$

$$\text{Normalised chimerism} = (N^+_{\text{donor}}(t) + N^-_{\text{donor}}(t))/(\chi N(t)),$$

$$\text{Ki67}^+ \text{ fraction within donor and host} = \frac{N^+_{\text{donor}}(t)}{N^+_{\text{donor}}(t) + N^-_{\text{donor}}(t)}, \quad \frac{N^+_{\text{host}}(t)}{N^+_{\text{host}}(t) + N^-_{\text{host}}(t)},$$

using the empirical descriptions of the size ($S(t)$) and Ki67$^+$ fraction ($\epsilon(t)$) of SP thymocytes, their direct precursors.

#### Density-dependent models

In these extensions of the neutral model, the rate of cell division $\rho$, or the rate of loss $\delta$ vary with the size of the naive T cell population. We explored models exhibiting density-dependence in either $\rho$ or $\delta$. Both models assume that all cells in the population follow the same rules of self-renewal and turnover, at any given time. We defined the density-dependence using Hill functions;

$$\rho(N) = \frac{\rho_0}{1 + \left(\frac{N^- + N^+}{\bar{C}^3}\right)}, \qquad \delta(N) = \frac{\delta_0}{1 + \left(\frac{N^- + N^+}{\bar{C}}\right)}, \tag{3}$$

where $\bar{C}$ and $a_0$ or $\delta_0$ were estimated from the model fits to the data.

#### RTE model

Here, we treated RTE and mature naive (MN) T cells separately, allowing them to have distinct rates of division ($\rho_R$ and $\rho_N$) and of loss ($\delta_R$ and $\delta_N$). We assume a constant rate of maturation of RTE ($\mu$). In this model, the expected residence time of cells in the RTE compartment is $1/(\delta_R + \mu)$, and a proportion $\mu/(\delta_R + \mu)$ of RTE survive to maturity. We solve the following equations for Ki67$^+$ and Ki67$^-$ cells with the RTE and MN compartments, which are identical for host- and donor-derived cells;

$$\frac{dR^-}{dt} = \alpha S(t)(1 - \epsilon) + \rho_R (2 R^-(t) + R^+(t)) - (\beta + \delta_R) R^+(t), \tag{4}$$

$$\frac{dR^+}{dt} = \alpha S(t)\epsilon + \beta R^+(t) - (\rho_R + \delta_R) R^-(t), \tag{5}$$

$$\frac{dN^-}{dt} = \mu R^-(t) + \rho_N (2 N^-(t) + N^+(t)) - (\beta + \delta_N) N^+(t), \tag{6}$$

$$\frac{dN^+}{dt} = \mu R^+(t) + \beta N^+(t) - (\rho_N + \delta_N)N^-(t). \tag{7}$$

## Age-dependent division and loss models

We aimed to model the population density of naive T cells $u(a, k, t)$, where $t$ is mouse age, $a$ is a cell's age (defined as the time since it or its ancestor left the thymus), and $k$ is a cell's level of Ki67 expression. We assume Ki67 expression reaches a maximal level of $k = 1$ immediately after cell division and decays exponentially at rate $\beta$. We assume that the *per capita* rates at which cells divide ($\rho$) and are lost ($\delta$) can be functions of mouse age $t$ and/or of cell age $a$. To model the evolution of $u(a, k, t)$, we extended an age-structured population PDE model described previously (**Hogan et al., 2015**; **Rane et al., 2018**) to include Ki67 expression.

**Initial conditions.** We assume that at some host age $t_0$, the population has size $N_0$ and has a cell-age distribution $\gamma(a)$, where $0 \leq a \leq t_0$ and $\int_0^{t_0} \gamma(a)\,da = 1$ (we assume all cells are of age $a = 0$ when they leave the thymus); and these cells have a distribution of levels of Ki67 expression $\psi(k)$, where $\int_0^1 \psi(k)dk = 1$ and $\psi(k) = 0$ for $k \notin (0, 1)$. Here for simplicity we are assuming no relation between Ki67 expression and cell age within the cells present at $t = t_0$, but one could easily extend this framework with a more general initial joint distribution $P(a, k)$. At times $t \geq t_0$, we assume that cells of age zero enter the naive pool from the thymus at rate $\theta(t)$ and with Ki67 distribution $\phi(k, t)$, where $\int_0^1 \phi(k, t)dk = 1$ for all $t$.

**Breaking the solution into cohorts of cells.** Our approach is to track separately the fates of cells that were present at mouse age $t_0$, who will all have age $a > t - t_0$ at some later time $t$; and the fates of those that were exported from the thymus at time $t_0$ or later, which will all have age $a < t - t_0$. We then add these to get the full population density $u(a, k, t)$. The master PDE for both populations combined is

$$\frac{\partial u}{\partial t} + \frac{\partial u}{\partial a} - \beta k \frac{\partial u}{\partial k} = -(\rho(a, t) + \delta(a, t))u(a, k, t) \tag{8}$$

with boundary conditions

$$u(a, k = 1, t) = 2\rho(a, t) \int_{k=0}^1 u(a, k, t)dk \quad \text{(Cell division)} \tag{9}$$

$$u(a, k, t = t_0) = N_0 \gamma(a)\psi(k) \quad \text{(Population present at host age } t_0) \tag{10}$$

$$u(a = 0, k, t) = \theta(t)\phi(k, t) \quad \text{(Influx of new cells from thymus)} \tag{11}$$

The first condition above derives from cell division; at any host age $t$, cells of age $a$ at time $t$ divide at rate $\rho(a, t)$ generating two cells of age $a$ with $k = 1$.

## Initial cohort

**Non-divided cells.** First consider those cells present at $t = t_0$ that have yet to divide; this population decreases in size with a *per capita* rate $\delta(a, t) + \rho(a, t)$, and follows **Equation 8** with the single boundary condition $u(a, k, t = t_0) = N_0\gamma(a)\psi(k)$. We solve this using the method of characteristics by identifying a variable $s$ such that

$$\frac{du}{ds} = -\Upsilon(a, t)u(a, k, t) = \frac{dt}{ds}\frac{\partial u}{\partial t} + \frac{da}{ds}\frac{\partial u}{\partial a} + \frac{dk}{ds}\frac{\partial u}{\partial k} \tag{12}$$

where for brevity we define $\Upsilon(a, t) = \rho(a, t) + \delta(a, t)$. Equating terms with **Equation 8** gives

$$dt/ds = 1, da/ds = 1 \implies s = t - t_0, \quad a = a_0 + t - t_0, \quad dk/ds = -\beta k \implies k = k_0 e^{-\beta(t - t_0)}. \tag{13}$$

Along the characteristic that starts at $(a_0, k_0, t_0)$, illustrated in **Appendix 1—figure 1A** below, the population density evolves as

$$\frac{d}{dt}u(a_0, k_0, t) = -\Upsilon(a, t)u(a_0, k_0, t) \tag{14}$$

$$= -\Upsilon(a, a - a_0 + t_0)u(a_0, k_0, t) \tag{15}$$

$$\implies u(a_0, k_0, t) = u(a_0, k_0, t_0) \exp\left(-\int_{x=a_0}^{a_0+t-t_0} \Upsilon(x, x - a_0 + t_0)dx\right). \tag{16}$$

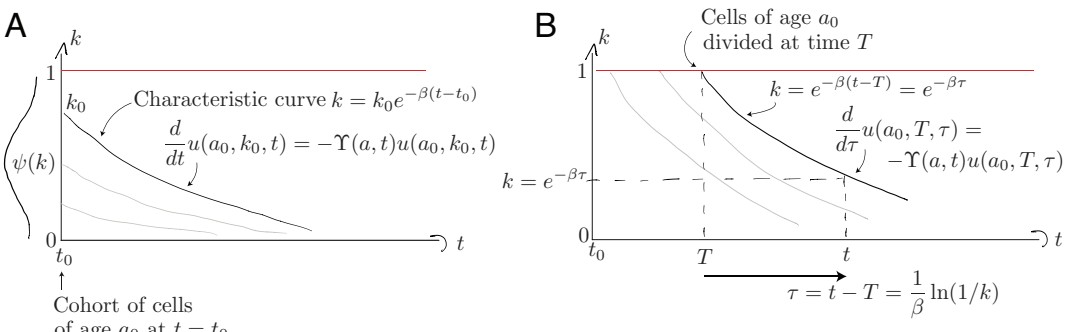

**Appendix 1—figure 1.** Characteristic curves for (**A**) the population of age $a_0$ present at $t = t_0$ and (**B**) the population who divided at time $T$ when they were of age $a_0$.

We know from *Equation 13* that $k_0 = ke^{\beta(t-t_0)}$, so

$$u(a_0, ke^{\beta(t-t_0)}, t) = u(a_0, ke^{\beta(t-t_0)}, t_0) \exp\left(-\int_{a_0}^{a_0+t-t_0} \Upsilon(x, x - a_0 + t_0)dx\right). \tag{17}$$

The population density $u(.)$ must then be transformed with a Jacobian to express it as a density over $k$ rather than over $ke^{\beta(t-t_0)}$. If $y = ke^{\beta(t-t_0)}$, then

$$u(a, k, t) = u(a_0, y, t) \,|\, dy/dk\,| = u(a_0, ke^{\beta(t-t_0)}, t)e^{\beta(t-t_0)}, \tag{18}$$

giving

$$\begin{aligned} u_{i,n}(a, k, t) &= u(a_0, ke^{\beta(t-t_0)}, t_0) \exp\left(\beta(t - t_0) - \int_{x=a_0}^{a_0+t-t_0} \Upsilon(x, x - a_0 + t_0)dx\right) \\ &= N_0\gamma(a - t + t_0)\psi(ke^{\beta(t-t_0)}) \exp\left(\beta(t - t_0) - \int_{x=a+t_0-t}^{a} \Upsilon(x, x - a + t) \, dx\right), \\ &\text{for } t \geq t_0, a \geq t - t_0, k \leq e^{-\beta(t-t_0)}. \end{aligned} \tag{19}$$

where the subscript $(i, n)$ denotes 'initial and non-divided'. *Equation 19* holds for $k \leq e^{-\beta(t-t_0)}$ because none of these cells have divided since time $t_0$ and their Ki67 expression is decaying $k = 1$.

   **Divided cells.** *Appendix 1—figure 1B* above illustrates the evolution of cells from the initial cohort who subsequently divide, each time resetting their Ki67 expression to $k = 1$. To follow these cohorts we solve *Equation 8* with the boundary condition describing the division of cells of age $a$ at host age $t$;

$$u(a, k = 1, t) = 2\rho(a, t) \int_{k=0}^{1} u(a, k, t) \, dk. \tag{20}$$

Characteristic curves are again of the general form $k = k_0e^{-\beta t}$, but we parameterise them differently to those described above, since these originate at $k = 1$ and not $t = t_0$. Cells with Ki67 expression $k$ must have divided a time $\tau = -(1/\beta)\ln(k)$ in the past. A convenient parameterisation is therefore

$$s = \tau, \quad a = a_0 + \tau, \quad k = e^{-\beta\tau}, \tag{21}$$

where the cells currently of age $a$ divided at time $T = t - \tau$ when they were aged $a_0 = a - \tau$. Along these curves,

$$u(a_0, T, \tau = (1/\beta) \ln(1/k)) = 2\rho(a_0, T) \left( \int_{x=0}^{1} u_i(a_0, x, T)\, dx \right) \exp\left( - \int_{x=a_0}^{a_0+\tau} \Upsilon(x, x - a_0 + T)\, dx \right) \quad (22)$$

where the exponential term represents the proportion of cells on this characteristic curve that divide again or die. It integrates a cell's experience from age $a_0$ at host age $T$, to age $a$ at host age $t = T + \tau$. Here, $u_i$ denotes the entire initial cohort, divided or undivided, integrated over all levels of Ki67 expression, whose cells of age $a_0$ fed the population at time $T$.

To convert this to a density $u_{i,d}(a, k, t)$, we use the Jacobian $d\tau/dk = 1/(\beta k)$. This gives

$$
\begin{aligned}
u_{i,d}(a, k, t) &= \frac{2\rho(a - \tau, t - \tau)}{\beta k} \left( \int_{x=0}^{1} u_i(a - \tau, x, t - \tau)\, dx \right) \exp\left( - \int_{x=a-\tau}^{a} \Upsilon(x, x - a + t)\, dx \right) \\
&= \frac{2\rho(a - \tau, t - \tau)}{\beta k} U_i(a - \tau, t - \tau) \exp\left( - \int_{x=a-\tau}^{a} \Upsilon(x, x - a + t)\, dx \right), \\
&\quad \text{for } \tau = (1/\beta) \ln(1/k),\ a \geq \tau,\ t - t_0 \geq \tau,\ k > e^{-\beta(t - t_0)},
\end{aligned}
\quad (23)
$$

where $U_i(a, t)$ is the population density of the initial cohort (divided or undivided) at age $a$ and time $t$, integrated over all $k$, which we will return to below.

## Cells that enter after $t = t_0$

A similar set of calculations applies for cells that subsequently enter the pool, at which point we define them to be of age zero. Again we partition these cells into those that will not divide and those that will. For the former, consider those that are of age $a \leq t - t_0$ at time $t$; that is, cells that entered later than $t_0$. We denote this population $u_{\theta,n}(a, k, t)$ where $k \leq e^{-\beta a}$. These cells are what remains of the cohort exported at time $t - a$, at age $a = 0$, of size $\theta(t - a)$ and with Ki67 distribution $\phi(k e^{\beta a}, t - a)$. These are then lost to death or division. By analogy with **Equation 19**, the Jacobian is $e^{\beta a}$ and these non-divided cells evolve as

$$u_{\theta,n}(a, k, t) = \theta(t - a)\phi\left( k e^{\beta a}, t - a \right) \exp\left( \beta a - \int_{x=0}^{a} \Upsilon(x, x - a + t)\, dx \right),$$

$$\text{for } t > t_0, a < (t - t_0), k \leq e^{-\beta a} \quad (24)$$

Here, cells are born with age zero and so the characteristics are $k = k_0 e^{-\beta a}$. Similarly for divided cells; by analogy with **Equation 23**,

$$u_{\theta,d}(a, k, t) = \frac{2\rho(a - \tau, t - \tau)}{\beta k} U_\theta(a - \tau, t - \tau) \times \exp\left( - \int_{x=a-\tau}^{a} \Upsilon(x, x - a + t)\, dx \right),$$

$$\text{for } \tau = (1/\beta) \ln(1/k), t \geq t_0, a \leq t - t_0, 0 \leq \tau \leq a \quad (25)$$

Here, $U_\theta(a, t)$ is the population density of cells of of age $a$ at time $t$, who entered the pool a time $\tau = t - a$ ago and may have divided or not.

## Solving the age-structured PDE only

To complete these solutions we need the population densities $U_i(a, t)$ and $U_\theta(a, t)$, ignoring Ki67 expression. These are straightforward;

$$U_i(a, t) = N_0 \gamma(a - t + t_0) \exp\left( - \int_{x=a-t+t_0}^{a} \lambda(x, x - a + t)\, dx \right), \quad \text{for } a \geq t - t_0, \quad (26)$$

$$U_\theta(a, t) = \theta(t - a) \exp\left( - \int_{x=0}^{a} \lambda(x, x - a + t)\, dx \right), \quad \text{for } a \leq t - t_0. \quad (27)$$

where for brevity $\lambda(a, t)$ is the net loss rate of cells of age $a$ at time $t$, which is $\delta(a, t) - \rho(a, t)$. The integral in **Equation 26** follows a cell whose age runs from $a - (t - t_0)$ to $a$, during which host age runs from $t - a$ to $t$. The integral in **Equation 27** follows a cell whose age runs from 0 to $a$, between

host ages of $t - a$ to $t$. The two solutions join at $a = t - t_0$; the influx at time $t_0$ must be the density of cells of age zero in the initial cohort; $\theta(t_0) = N_0 \gamma(0)$.

Therefore, to obtain the total solution $u(a, k, t) = u_{i,n} + u_{i,d} + u_{\theta,n} + u_{\theta,d}$, we add *Equations 19, 23, 24 and 25*, using the solutions for the age-structured model given in *Equations 26 and 27*.

We can connect this solution to gated flow cytometry data by partitioning the population into high and low Ki67 expression. We define Ki67$^+$ cells to be those which divided no more than a time $1/\beta$ ago, which corresponds to a cut-off of $k = 1/e$. Therefore, the numbers of Ki67 positive and negative cells at time $t$ are

$$N^+(t) = \int_{k=1/e}^{1} \int_{a=0}^{a_{\max}} u(a, k, t) \, dk \, da, \qquad N^-(t) = \int_{k=0}^{1/e} \int_{a=0}^{a_{\max}} u(a, k, t) \, dk \, da. \tag{28}$$

## Appendix 2

### Constructing empirical descriptions of the dynamics of mature SP thymocytes over the mouse lifespan

We assumed that the rate of export of new naive T cells from the thymus is proportional to the numbers of single positive (SP4 and SP8) thymocytes (*Berzins et al., 1998*). The total numbers of SP thymocytes increase rapidly up to 6–7 weeks of age and then drop gradually over time. We used the following empirical descriptor function to capture the dynamics of thymic SP cells varying with mouse age;

$$S(t) = S(0) + At^n \frac{(1-t^q)}{B^q+t^q}. \tag{29}$$

We estimated the parameters $S(0)$, $A$, $n$ and $q$ by fitting *Equation 29* to the log-transformed data from wild-type mice bred in the same facility as the busulfan chimeras (*Appendix 2—figure 1A*). The rate of thymic export is then $\theta(t) = \alpha S(t)$, with the constant $\alpha$ estimated when fitting models to the busulfan chimera data.

We modelled the Ki67$^+$ fraction within SP4 and SP8 thymocytes using the form

$$\epsilon(t) = \epsilon_0 + e^{-\epsilon_f(t+C)}, \tag{30}$$

and estimated $\epsilon_0$, $\epsilon_f$ and $C$ by fitting this function to the logit-transformed proportions of cells that were Ki67$^+$ (*Appendix 2—figure 1B*).

When modelling data from the busulfan chimeras, we assumed that the total output from the thymus at any time is identical to that in age-matched wild type mice, but is split between donor and host cells according to the chimerism $\chi$ at the DP1 stage of thymic development:

$$\theta_{\text{donor}}(t) = \chi\theta(t); \quad \theta_{\text{host}}(t) = (1-\chi)\theta(t). \tag{31}$$

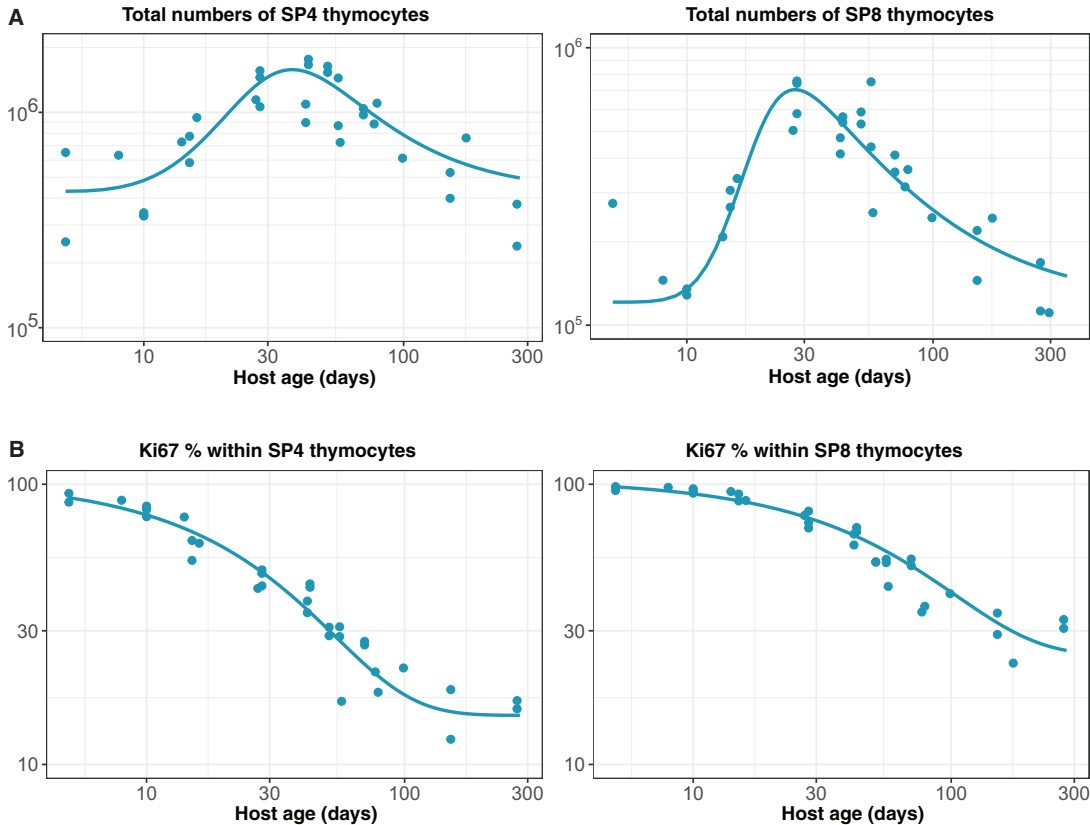

**Appendix 2—figure 1.** Empirical descriptions of the dynamics of the numbers and Ki67 expression of late-stage thymocytes. These curves (defined above) were used as inputs to models of the data from adult busulfan chimeric mice.

# Appendix 3

## Model fitting and selection criteria

Each of the models described in Appendix 1 was fitted simultaneously to four sets of observations – cell counts, normalised chimerism, and the proportions of cells expressing Ki67 within donor and host naive T cells. To normalise residuals, cell counts were log transformed; and normalised chimerism and Ki67$^+$ fractions were logit- and arcsine square root-transformed, respectively. We used a Bayesian inference approach to estimate the model parameters and the errors associated with the measurements in each dataset. Model definitions, the prior distribution of parameters and the likelihood definitions were encoded in the *Stan* language and are available at this Github repository, and models were fitted using the Hamiltonian Monte Carlo algorithm (***Team Stan Development, 2022***).

We compared the support for models using the leave-one-out (LOO) cross validation method. The expected log point-wise predictive density (elpd) of the model $M_k$, which is a measure of its out-of-sample prediction accuracy (***Vehtari et al., 2016***), can be estimated using LOO as

$$\widehat{\mathrm{elpd}_{\mathrm{loo}}^{k}} = \sum_{i=1}^{n} \mathrm{elpd}_{\mathrm{loo,\,i}}^{k} = \sum_{i=1}^{n} \log(P(y_i \,|\, y_{-i}, M_k)), \qquad (32)$$

where $P(y_i \,|\, y_{-i}, M_k)$ is the likelihood of observing $y_i$ given model $M_k$ fitted on the data with observation $i$ excluded. The elpd estimate is the sum of $n$ independent components, and so its standard error is

$$\mathrm{se}(\widehat{\mathrm{elpd}_{\mathrm{loo}}^{k}}) = \sqrt{\sum_{i=1}^{n} (\mathrm{elpd}_{\mathrm{loo,\,i}}^{k} - \mathrm{elpd}_{\mathrm{loo}}^{k}/n)^2}. \qquad (33)$$

We calculated the elpd estimate using the *loo-2.0* package in the *Rstan* library, which employs Pareto smoothed importance sampling (PSIS) (***Vehtari et al., 2015***) to approximate LOO cross validation. The input for this process is an array of joint likelihoods evaluated at draws from the posterior parameter distributions. The estimates of elpd and its standard error were used to rank models using the Pseudo-Bayesian model averaging (BMA) method (***Yao et al., 2018***), which is analogous to using Akaike's Information Criterion (AIC) to calculate model weights ***Akaike, 1978***; ***Burnham and Anderson, 2002***; ***Wagenmakers and Farrell, 2004***, given by

$$W_k = \frac{\exp(\widehat{\mathrm{elpd}_{\mathrm{loo}}^{k}} - \frac{1}{2}\mathrm{se}(\widehat{\mathrm{elpd}_{\mathrm{loo}}^{k}}))}{\sum_{k=1}^{K} \exp\left(\widehat{\mathrm{elpd}_{\mathrm{loo}}^{k}} - \frac{1}{2}\mathrm{se}(\widehat{\mathrm{elpd}_{\mathrm{loo}}^{k}})\right)}. \qquad (34)$$

An intuitive interpretation of this formulation is that models are given a greater weight for both high total out-of-sample prediction accuracy and for evenness of this accuracy across the set of observations. These Pseudo-BMA model weights were calculated using the *loo-2.0* package and are reported in ***Table 1*** in the text.

## Appendix 4

### Predicting the kinetics of loss of adoptively co-transferred RTE and mature naive T cells

To compare the dynamics of recent thymic emigrants (RTE) and mature naive (MN) T cells, we simulated the co-transfer experiment described in *Houston et al., 2011*, using age-dependent loss and division models with parameters derived from fitting to the busulfan chimera data.

Houston et al. isolated RTE from 5- to 9-week-old RAG2-GFP reporter mice, and MN T cells from a mixture of thymectomised WT and RAG2-GFP mice aged greater than 12 weeks. These cells were co-transferred to WT recipient mice.

To predict the subsequent kinetics of the transferred populations with our models, we needed to define their age-distributions at the time of transfer. This involved an approximation, given the uncertainty in the ages of the donor mice. We assumed that the RTE and MN populations were sampled from mice of age 5 weeks and 20 weeks. We then solved the age-structured PDE only, without tracking Ki67 expression (Appendix 1, *Equations 26 and 27*), until age 5 weeks or 20 weeks and then simulated the process of transfer by setting thymic influx to zero and observing the decay of the population. We then enumerated RTE and MN cells integrating the appropriate regions of the cell age distribution at the specified times post-transfer, defining RTE as cells with a post-thymic age of 10 days or less, and MN cells being older than 28 days.

## Appendix 5

### Predicting RTE dynamics in Rag/Ki67 dual reporter mice

In Rag$^{GFP}$ Ki67$^{RFP}$ reporter mice, RTE are identified based on the transient expression of GFP, which we assume decays with first order kinetics. Our models estimated very low rates of division among naive CD4 and CD8 T cells, such that any dilution of GFP through division is minimal. There is therefore a simple and direct correlation between GFP expression $f$ and cell age $a$ within naive T cells, which we used to predict the fractions of GFP$^+$ cells ($F$) within the naive compartment.

Using the favoured age-structured models described in Appendix 1, the age distribution of the naive T cell pool at mouse age $t$ is given by **Equations 26 and 27** as $U(a,t) = U_i(a,t) + U_\theta(a,t)$. The fraction of cells that are GFP$^+$ is then

$$F = \frac{\int_0^{\bar{a}} U(a,t)}{\int_0^t U(a,t)},$$

(35)

where $\bar{a}$ is the (unknown) time required for cells to transition from GFP$^+$ to GFP$^-$. Implicit in this calculation is the assumption that GFP levels are similar in Ki67$^-$ and Ki67$^+$ RTE; mature SP cells are indeed very bright for GFP with minimal differences when stratified by Ki67 expression (data not shown). We then used the parameters derived from busulfan chimera data to generate $U(a,t)$, and estimated $\bar{a}$ by fitting **Equation 35** to the timecourse of the GFP$^+$ fraction within naive T cells observed in Rag/Ki67 dual reporter mice. The fits are shown in **Figure 4C and D**, red lines in the leftmost panels. We then generated the predicted timecourses of the GFP$^+$ Ki67$^+$ and GFP$^+$ Ki67$^-$ fractions by multiplying $F$ with the predicted Ki67$^+$ and Ki67$^-$ fractions that we derived by running the age-dependent loss model from the age of the youngest GFP/Ki67 reporter mouse (11 days).

## Appendix 6

### Extending models back to near-birth to predict the dynamics of naive T cells in neonates

In order to model the busulfan chimera data, in which mice underwent BMT at different ages, we needed to define the healthy dynamics of naive T cells. We took the approach of evolving the naive T cell pool from 1 day of age assuming that kinetic parameters were constant across the lifespan. In this way, every mouse began from the same 'baseline' state just after birth, and varied only in its age at BMT and in the level of stable chimerism within the bone marrow and thymus. The free parameters for this aspect of the model were then the (unknown) numbers of Ki67$^+$ and Ki67$^-$ naive CD4 or CD8 T cells in a 1-day-old mouse, $N_0^+$ and $N_0^-$ respectively. For each model, we then calculated the predicted numbers of host-derived Ki67$^+$ and Ki67$^-$ cells at the age of BMT ($t_{\text{BMT}}$) by simulating forward from this initial condition, allowing for the continued influx of Ki67$^+$ and Ki67$^-$ cells from the thymus as described in Appendix 2.

In cell-age dependent models, we also defined the Ki67 distribution within the pre-existing naive T cells (initial cohort) and among cells subsequently exported from the thymus emigrants. We also assumed a uniform distribution of cell ages ($\gamma(a) = 1$) in 1-day-old mice.

**Initial cohort:** Naive T cells present at mouse age $t_0 = 1$ day are enriched with Ki67$^+$ cells (*Figure 4A and B*), and we defined the distribution of their normalised Ki67 expression $k$ over (0,1] to be $\psi(k) = e^k/(e^k - 1)$. We assume Ki67 is lost exponentially at a constant rate $\beta$, such that at a time $s$ after division $k(s) = e^{-\beta s}$. Our results were not sensitive to the form of $\psi(k)$ since it 'washes out' on a timescale of $1/\beta = 3.5$ days.

We define the cut-off that separates Ki67$^+$ from Ki67$^-$ cells to be $\bar{k} = 1/e$, such that cells spend a time $1/\beta$ as Ki67$^+$. The fraction of Ki67$^+$ cells in the initial cohort is then obtained by integrating the initial Ki67 distribution $\psi(k)$ from $k = \bar{k}$ to $k = 1$, yielding

$$N_0^+ = N_0 \int_{\bar{k}}^1 \psi(k)dk \tag{36}$$

$$N_0^- = N_0 \left(1 - \int_{\bar{k}}^1 \psi(k)dk\right), \tag{37}$$

where $N_0$ is the total number of naive CD4 or CD8 T cells present in a 1-day-old mouse, and was a free parameter in the models.

#### Naive T cells that subsequently enter the periphery:

The Ki67$^+$ fraction within SP thymocytes ($\epsilon(t)$; *Equation 30*) varies with time, which implies that the distribution of Ki67 expression within naive T cells of age zero (i.e., just exported from the thymus) also change with time. We define this distribution to be $\phi(k, t)$, such that the Ki67$^+$ fraction among new RTE at mouse age $t$ is

$$\int_{\bar{k}}^1 \phi(k, t)dk = \epsilon(t) \tag{38}$$

We then defined an empirical step function $\phi(k, t)$ to be consistent with this relationship;

$$\phi(k, t) = \begin{cases} (1 - \epsilon(t))/\bar{k} & k \in [0, \bar{k}], \\ \epsilon(t)/(1 - \bar{k}) & k \in [\bar{k}, 1]. \end{cases} \tag{39}$$

To generate the predicted numbers and Ki67$^+$ fractions of naive CD4 and CD8 T cells from age 5 days onwards (*Figure 4*), it was then straightforward to plot the model fits derived from the adult busulfan chimeric mice; note that none of the data derived from the younger, wild-type mice were used in the fitting.

## Appendix 7

### Hierarchical modelling of naive CD8 T cell timestamping data

The data from *Reynaldi et al., 2019*, shown in *Figure 5*, comprised longitudinal samples drawn from animals in five different age groups who were each treated with pulses of tamoxifen to label cohorts of CD8 T cells leaving the thymus. To use these data to estimate how cell loss rates vary as a function of both cell and host age, we took a hierarchical modelling approach. We allowed for animal and/or group-level variation in the initial numbers of RFP labelled cells in each animal ($N_0$) and in their initial loss rate (that is, the instantaneous net loss rate of cells of age zero, just exported from the thymus). We began by modeling the kinetics of labelled cells with the assumption that their net loss rate $\lambda$ varies with their post-thymic age $a$ as $\lambda(a) = \lambda_0 e^{-\gamma a}$. The population density of cells of age $a$ at mouse age $t$, $N(a, t)$, then obeys

$$\frac{\partial N}{\partial t} + \frac{\partial N}{\partial a} = -\lambda(a)N(t, a),$$

(40)

with the boundary condition $N(T, a) = N_0 \, \delta(a)$, where $T$ is the mouse age at the time of treatment and $\delta(.)$ is the Dirac delta function. For mouse $i$, in age group $j$, at timepoint $k$, the observed cell numbers are $y_{ijk}$, where

$$\ln y_{ijk} \sim \mathcal{N}(z_{ijk}, \sigma) \qquad \text{(Likelihood)}$$

$$z_{ijk} = f(t_k, N_{0,ij}, \lambda_{0,ij}, \gamma) \qquad \text{(Model)}$$

(41)

We considered models in which the initial cell numbers $N_0$ and initial loss rate $\lambda_0$ for each mouse were either drawn from a single parent distribution or from distributions with means and variances specific to each age group. We defined priors for $\mu_N$, $\sigma_N$, $\mu_\lambda$, $\sigma_\lambda$ and $\gamma$ and fitted permutations of the hierarchical age-structured model to the time courses of labelled CD8 T cell numbers (*Appendix 7—table 1*). The best-fitting model exhibited group-specific values of the initial RTE loss rate $\lambda_0$, and variation in the initial numbers of labelled cells ($N_0$) across mice, likely deriving from variations in the efficiency of tamoxifen-driven labelling.

**Appendix 7—table 1.** Comparing support for hierarchical age-structured models of the data from *Reynaldi et al., 2019*.

| Model | Initial numbers | Net loss rate at age 0 | ΔLOO-IC | Weight % |
|---|---|---|---|---|
| $N_0$ varying at animal level | $N_{0,i} \sim \mathcal{N}(\mu_N, \sigma_N)$ | constant | 316 | 0.0 |
| $N_0$ varying at animal level; $\lambda_0$ varying at group level | $N_{0,i} \sim \mathcal{N}(\mu_N, \sigma_N)$ | $\lambda_{0,j} \sim \mathcal{N}(\mu_\lambda, \sigma_\lambda)$ | 0.0 | 100 |
| $N_0$ varying at animal level; $\lambda_0$ varying at animal level | $N_{0,i} \sim \mathcal{N}(\mu_N, \sigma_N)$ | $\lambda_{0,i} \sim \mathcal{N}(\mu_\lambda, \sigma_\lambda)$ | 73 | 0.0 |
| $N_0$ varying at group level; $\lambda_0$ varying at animal level | $N_{0,j} \sim \mathcal{N}(\mu_N, \sigma_N)$ | $\lambda_{0,i} \sim \mathcal{N}(\mu_\lambda, \sigma_\lambda)$ | 313 | 0.0 |

We then generated an explicit, empirical description of the variation in $\lambda_0$ with mouse age $t$, using the group-specific estimates of $\lambda_0$ from the best-fitting hierarchical model. We used the following model of the net loss rate of cells of age $a$ at time $t$, with estimated parameters $\lambda_h, Q, q$ and $\gamma$;

$$\lambda(t, a) = \lambda_0(t) \, e^{-\gamma \, a} = \lambda_h \left( 1 + \frac{Q}{1 + (t/q)^5} \right) e^{-\gamma \, a}.$$

(42)

## Appendix 8

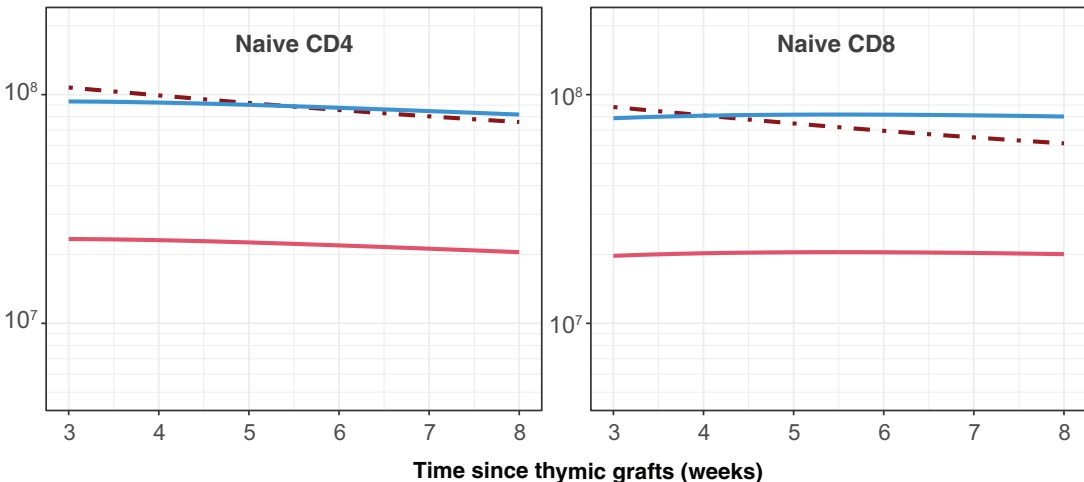

**Appendix 8—figure 1.** Simulating the outcome of transplanting 6 additional thymi, as described by *Berzins et al., 1999*. The change in numbers of naive CD4 and CD8 T cells is equivalent to 3 weeks of thymic output.

