## [Editor Report]

This paper challenges the widely held view that the number of naive T cells in our body is regulated through homeostatic feedback mechanisms, meaning that cells divide more frequently – or live longer – when cell numbers are low. The arguments in favor of this homeostatic regulation rely on cross-sectional data, which fail to distinguish between the effects of host age, cell age, and cell numbers, each of which separately may influence the dynamics of cells. Using a set of mathematical models and experimental data sets, this paper manages to tear these factors apart and reports that in mice there is no feedback regulation of naive T cell numbers.

---

## [Decision Letter]

**Decision letter after peer review:**

Thank you for submitting your article "Towards a unified model of naive T cell dynamics across the lifespan" for consideration by *eLife*. Your article has been reviewed by 2 peer reviewers, and the evaluation has been overseen by a Reviewing Editor and Tadatsugu Taniguchi as the Senior Editor. The following individual involved in the review of your submission has agreed to reveal their identity: Jose Borghans (Reviewer #1).

Essential revisions:

The reviewers have noted several elements of the work that require clarification:

1) Expansion of the explanation regarding several points of the model.

2) Attention to revisions of the figures and figure legends and provision of additional qualifications of the models (eg. quantitative validation such as correlation coefficients).

3) Consider points raised regarding additional approaches in modelling/testing the data.

*Reviewer #1 (Recommendations for the authors):*

– Some sentences/claims in the abstract are very general; please specify that this is the situation in (clean) mice.

– I found Figure S2 very helpful. I would advise incorporating it as the first main figure of the paper.

– If I understood correctly, in Figure 1B, the authors have plotted the combined data of mice who underwent BMT at different ages (between 7 and 25 weeks of age). I'm wondering whether the age at BMT may have an influence on the measurements. It may be interesting to use different labels to specify the age at BMT.

– Figure 2: Please plot the fits for all the models that were tested (even if only in supplemental information). That would help the reader judge visually how bad the fits of the alternative models were and would show in which ranges those models fit poorly. The lower relative weights of the alternative models suggest that the fits of the alternative models were really bad, but I would like to see that also visually.

– In Figure 2, I noticed that in the Ki67 panels, the early data points describing the donor cells are consistently missed by the best fits. Do the authors know why this is the case? Please explain. In the last Ki67 panel (11-25 weeks), on the other hand, the donor data seem to be almost overfitted.

– On page 4, the authors state that "A prediction of any progressive increase in cell division rates with age is also inconsistent with observations of the frequencies of T cell receptor excision circles (TRECs) in adult mice." The authors also report, however, that the best fit of the naive CD8 T cell data is obtained for an age-dependent division model in which the T cell division rate increases only very slightly with age. I'm not convinced that the situation would be inconsistent with the TREC data.

– On the same page, it is stated "This model was very similar to a neutral, homogeneous model and predicted that the normalised donor fraction approaches 1 in aged mice. This conclusion contradicts findings from our own and others' studies that demonstrated that models assuming homogeneity in naive CD8 T cells failed to capture their dynamics in adult and aged mice (2-20 months old)." The authors explain this as follows: "The statistical support for the age-dependent division model here may derive from the relatively sparse observations in aged mice in this dataset, which define the asymptotic replacement fraction." I'm afraid I miss the point; I fail to understand this explanation.

– I am struggling to understand intuitively why the age-dependent division model performs so poorly on the data of Figure 3B, while the model performs really well on the data of Figure 2. If the proliferation rate were to be fitted again on the basis of Figure 3, would the fit to the data of Figure 2 become very bad? What happens if both data sets would be fitted simultaneously?

– Discussion: It would be very much appreciated if the authors could discuss their view on the translation of these findings to the human and the dirtier mouse situation.

– In the Discussion it is mentioned that "Naive T cells proliferate under severely lymphopenic conditions in mice but acquire a memory-like phenotype. There is some evidence that this process occurs in healthy neonatal mice, suggesting that they are lymphopenic to some degree." I wanted to note that in Hoeven et al. (ref 22), we observed that this increased transition of RTE to the memory pool even occurs in the lympho-replete situation, suggesting that it's related to the RTE nature of the cells rather than to the cell numbers in the host. The authors may want to consider adding this to their discussion.

*Reviewer #2 (Recommendations for the authors):*

I think it would have been helpful to have seen quantitative descriptions of model fits (e.g. R^2^ values) in the main body or captions, which I think would help substantiate some of the remarks.

For the qualitative fit in Figure 3, perhaps some speculative statement (if you have one) might be worth making to explain the quantitative discrepancy?

I'd suggest that all of the figures and captions need to be re-read carefully for errors. It might be excessive, but in all captions (or recorded in SI if it is too unattractive in the main text) it would have been useful to know the number of samples/mice in each panel. These are also absent from the main text.

Figure 1 is missing caption entries for panels C and D.

Figure 2. Reporting goodness of fit statistics in the caption might be informative.

Figure 3. Perhaps the top two panels should be A and the bottom two B, with appropriate classification in the caption? As this is an important figure, I think it would be worth expanding on the simulation, even with some brief text, as there appears to be none presently.

In Figure 5B, what are the units for the y-axis?

In Figure 6B, if these are paired measurements, it might be appropriate to report the Spearman correlation coefficient to quantitatively support the (obvious anyway) relationship between SP and Naive.

Check that BMT, bone marrow transplant, and SP, for single positive, are defined the first time they are used.

Did you consider fitting your models to a bootstrapped version of your data to get estimates on the influence of experimental variability? That might, e.g., have given an indication as to the strength of belief in the CD8^+^ age-dependent division model in the first section.

---

## [Author Response]

Essential revisions:The reviewers have noted several elements of the work that require clarification:1) Expansion of the explanation regarding several points of the model.2) Attention to revisions of the figures and figure legends and provision of additional qualifications of the models (eg. quantitative validation such as correlation coefficients).3) Consider points raised regarding additional approaches in modelling/testing the data.

We’re really grateful for the care both reviewers took in understanding and critiquing our manuscript. We feel that addressing their comments has helped us to improve it substantially.

Reviewer #1 (Recommendations for the authors):– Some sentences/claims in the abstract are very general; please specify that this is the situation in (clean) mice.

We now clarify in several places that we are dealing with mice.

– I found Figure S2 very helpful. I would advise incorporating it as the first main figure of the paper.

This is a good idea – we’ve moved the model schematics to Figure 1 – see response below, too.

– If I understood correctly, in Figure 1B, the authors have plotted the combined data of mice who underwent BMT at different ages (between 7 and 25 weeks of age). I'm wondering whether the age at BMT may have an influence on the measurements. It may be interesting to use different labels to specify the age at BMT.

Yes, Figure 1 didn’t present the data in the clearest way. We were indeed careful to take both mouse age and age at BMT explicitly into account in the modelling. Specifically, we used the empirical description of single-positive thymocyte numbers with mouse age as a surrogate for thymus output, and used the appropriate region of this function when modeling host and donor cell dynamics in each mouse, with its particular age at BMT.

The data were all shown more clearly in Figure 2, where mouse age and age at BMT are presented more logically. So we’ve removed the raw data from Figure 1, and inserted the model sketches here instead.

– Figure 2: Please plot the fits for all the models that were tested (even if only in supplemental information). That would help the reader judge visually how bad the fits of the alternative models were and would show in which ranges those models fit poorly. The lower relative weights of the alternative models suggest that the fits of the alternative models were really bad, but I would like to see that also visually.

Good idea – we’ve now added these to the SI. None of the alternative models look atrocious, even when viewed under the transformations they were fitted on (e.g. the logit-scale for fractional measurements). When fitting to so many observations and different timecourses simultaneously, we find visual inspection doesn’t often square neatly with likelihood measures.

This problem (of strong statistical leanings vs. visually similar fits) highlights one reservation we have about relying on information criteria alone. The validity of any likelihood-based measure depends on the assumptions underlying it – in particular, the error distributions (e.g. the assumption of lognormal noise in cell numbers). While we do our best to transform data to satisfy these error models, departures will occur. For this reason we used these model weights as a guide only, and focused instead on out-of-sample prediction.

– In Figure 2, I noticed that in the Ki67 panels, the early data points describing the donor cells are consistently missed by the best fits. Do the authors know why this is the case? Please explain. In the last Ki67 panel (11-25 weeks), on the other hand, the donor data seem to be almost overfitted.

Thank you for picking this up! This was actually just an error in plotting the Ki67 predictions. They involve a subtraction step (age of mouse minus age at BMT) and this was done correctly for the fits to cell numbers and donor chimerism, but was incorrect for the Ki67 panels – so the fitted curves were slightly offset. We’ve corrected this.

– On page 4, the authors state that "A prediction of any progressive increase in cell division rates with age is also inconsistent with observations of the frequencies of T cell receptor excision circles (TRECs) in adult mice." The authors also report, however, that the best fit of the naive CD8 T cell data is obtained for an age-dependent division model in which the T cell division rate increases only very slightly with age. I'm not convinced that the situation would be inconsistent with the TREC data.

Thanks, our logic was incorrect here – we’ve removed this argument.

– On the same page, it is stated "This model was very similar to a neutral, homogeneous model and predicted that the normalised donor fraction approaches 1 in aged mice. This conclusion contradicts findings from our own and others' studies that demonstrated that models assuming homogeneity in naive CD8 T cells failed to capture their dynamics in adult and aged mice (2-20 months old)." The authors explain this as follows: "The statistical support for the age-dependent division model here may derive from the relatively sparse observations in aged mice in this dataset, which define the asymptotic replacement fraction." I'm afraid I miss the point; I fail to understand this explanation.

We’ve tried to clarify. Essentially it was about information. The cell-age-dependent kinetics models are constrained most tightly by the donor replacement curve, whose asymptote is rather noisy for CD8s. With these data, therefore, our intuition is that we are not able to reliably discriminate between age-dependent loss and division for CD8s based on information criteria alone.

– I am struggling to understand intuitively why the age-dependent division model performs so poorly on the data of Figure 3B, while the model performs really well on the data of Figure 2. If the proliferation rate were to be fitted again on the basis of Figure 3, would the fit to the data of Figure 2 become very bad? What happens if both data sets would be fitted simultaneously?

This comment was helpful – we now explain that Figure 3 just serves as confirmation of strong cell-age effects for both CD4 and CD8 T cells; therefore, it excludes homogeneous models and lends weight to our decision to reject the fitted age-dependent division model for CD8s.

The age-dependent division models do poorly in Figure 3 because, for both CD4 and CD8 T cells, the predicted magnitudes of the division rate and its rate of increase with cell age are both very small when they are fitted to the data in Figure 2. This makes both the CD4 and CD8 age-dependent division models similar to neutral models with constant rates of division and loss, with no difference in the loss rates of RTE vs mature naive T cells.

Direct fitting of the age-dependent division model to the data in Figure 3 is difficult because the age of the mice in that experiment were not specified precisely, the MN cell donors were a mix of WT and thymectomised mice, and there is also ambiguity in the mapping of GFP-positivity to cell age, needed to define an RTE. We have extended our discussion of this point.

We went back-and-forth on the issue of fitting the models to everything at once vs fitting on data from one set of experiments and using out-of-sample prediction to challenge the models. Batch effects can be quite significant — for example, when comparing the dynamics of endogenous cells in Figure 2 to those of adoptively transferred cells in Figure 3; different housing facilities, reagents, and so on. So the correct way to fit simultaneously to observations from different labs and experimental systems would be to use hierarchical models. That would leave only information criteria with which to weigh support for different models, which as we describe above we find a little unsatisfactory. So in this study we tried to use out-of-sample prediction — even if just describing trends in new datasets — as a means of evaluating the models.

– Discussion: It would be very much appreciated if the authors could discuss their view on the translation of these findings to the human and the dirtier mouse situation.

Yes, this is important. We have added a section to the Discussion regarding self-renewal clonality, and added comments on T cell behaviour in humans and ‘dirty’ mice at other points in the text.

– In the Discussion it is mentioned that "Naive T cells proliferate under severely lymphopenic conditions in mice but acquire a memory-like phenotype. There is some evidence that this process occurs in healthy neonatal mice, suggesting that they are lymphopenic to some degree." I wanted to note that in Hoeven et al. (ref 22), we observed that this increased transition of RTE to the memory pool even occurs in the lympho-replete situation, suggesting that it's related to the RTE nature of the cells rather than to the cell numbers in the host. The authors may want to consider adding this to their discussion.

Thank you – we have added this point to our discussion and reworded that argument. (Your result also suggests to us that the continuous recruitment of new cells into memory-phenotype populations in the absence of infection likely derives substantially from RTE, rather than from random recruitment from the naive pool as a whole). Your study also reminded us to discuss the CD4 co-transfer experiment in Dong et al.. We also have difficulty reconciling their results with other studies, but we speculate that it may derive from the step of labelling one of the transferred populations (the bulk naive cells, which were lost fastest) with CFSE, but not the other.

Reviewer #2 (Recommendations for the authors):I think it would have been helpful to have seen quantitative descriptions of model fits (e.g. R^2^ values) in the main body or captions, which I think would help substantiate some of the remarks.

We agree that the measures of model fit were not prominent enough. We moved the table showing model weights from the SI to the main text, and we have added a more complete description of how these weights are calculated (they reflect each models’ average out-of-sample prediction error).

R^2^ is just a measure of closeness of fit and tends to 1 for overfitted data – so we don’t feel it is so helpful here where our goal to trade off goodness of fit vs. model complexity.

For the qualitative fit in Figure 3, perhaps some speculative statement (if you have one) might be worth making to explain the quantitative discrepancy?

We did speculate in the discussion, but have expanded on it. We believe the disparity stems from a combination of the unknown ages of the mice used in their experiments (in these models, we need to know the full age-distribution in order to calculate the net loss rate); potential effects of cell manipulation (adoptive transfer vs dynamics of endogenous cells); and uncertainty in the age cut-off defining RTE.

I'd suggest that all of the figures and captions need to be re-read carefully for errors. It might be excessive, but in all captions (or recorded in SI if it is too unattractive in the main text) it would have been useful to know the number of samples/mice in each panel. These are also absent from the main text.Figure 1 is missing caption entries for panels C and D.

Thanks – we’ve gone through all of the captions, made corrections and added sample sizes.

Figure 2. Reporting goodness of fit statistics in the caption might be informative.

See above for our comment re: goodness of fit.

Figure 3. Perhaps the top two panels should be A and the bottom two B, with appropriate classification in the caption? As this is an important figure, I think it would be worth expanding on the simulation, even with some brief text, as there appears to be none presently.

We reframed the discussion of Figure 3 and have actually compressed the figure a little — we feel it’s clearer now. We have also added a new section in the SI that details how the simulations were performed.

In Figure 5B, what are the units for the y-axis?

Net loss rate (loss rate – division rate) – this is the inverse of the expected clonal lifespan. We’ve added this. Also for panel 5A.

In Figure 6B, if these are paired measurements, it might be appropriate to report the Spearman correlation coefficient to quantitatively support the (obvious anyway) relationship between SP and Naive.

This is a good idea – we now quote the Spearman rank correlation coefficients.

Check that BMT, bone marrow transplant, and SP, for single positive, are defined the first time they are used.

Thank you for picking these up.

Did you consider fitting your models to a bootstrapped version of your data to get estimates on the influence of experimental variability? That might, e.g., have given an indication as to the strength of belief in the CD8^+^ age-dependent division model in the first section.

We are using a Bayesian fitting approach – the envelopes we show on the model predictions are essentially an analog of the envelopes you would get by bootstrapping the data and refitting (they represent the 2.5 and 97.5 percentiles of the model prediction taken across the posterior distributions of all the model parameters).